# Learned Sorted Table Search and Static Indexes in Small-Space Data Models †

Domenico Amato ⬤, Raffaele Giancarlo *⬤ and Giosué Lo Bosco ⬤

Dipartimento di Matematica e Informatica, Universitá degli Studi di Palermo, 90123 Palermo, Italy;
* Correspondence: raffaele.giancarlo@unipa.it
† An extended abstract related to this paper has been presented at the 20th International Conference of the Italian Association for Artificial Intelligence (AixIA 2021), Milano, Italy, 1–3 December 2021. The proceedings are Lecture Notes in Computer Science Vol. 13196, by Springer.

**Abstract:** Machine-learning techniques, properly combined with data structures, have resulted in Learned Static Indexes, innovative and powerful tools that speed up Binary Searches with the use of additional space with respect to the table being searched into. Such space is devoted to the machine-learning models. Although in their infancy, these are methodologically and practically important, due to the pervasiveness of Sorted Table Search procedures. In modern applications, model space is a key factor, and a major open question concerning this area is to assess to what extent one can enjoy the speeding up of Binary Searches achieved by Learned Indexes while using constant or nearly constant-space models. In this paper, we investigate the mentioned question by (a) introducing two new models, i.e., the Learned *k*-ary Search Model and the Synoptic Recursive Model Index; and (b) systematically exploring the time–space trade-offs of a hierarchy of existing models, i.e., the ones in the reference software platform *Searching on Sorted Data*, together with the new ones proposed here. We document a novel and rather complex time–space trade-off picture, which is informative for users as well as designers of Learned Indexing data structures. By adhering to and extending the current benchmarking methodology, we experimentally show that the Learned *k*-ary Search Model is competitive in time with respect to Binary Search in constant additional space. Our second model, together with the bi-criteria Piece-wise Geometric Model Index, can achieve speeding up of Binary Search with a model space of 0.05% more than the one taken by the table, thereby, being competitive in terms of the time–space trade-off with existing proposals. The Synoptic Recursive Model Index and the bi-criteria Piece-wise Geometric Model complement each other quite well across the various levels of the internal memory hierarchy. Finally, our findings stimulate research in this area since they highlight the need for further studies regarding the time–space relation in Learned Indexes.

**Keywords:** sorted table search; database management; learned indexes; machine learning

## 1. Introduction

Fundamental data structures, such as Hash Tables and Balanced Search Trees, are not only very useful in a broad range of application domains but are also so fundamental for computer science to be part of widely adopted textbooks in the discipline, e.g., [1]. With the aim of obtaining time and space performance improvements, an emerging trend is to combine machine-learning techniques with data-structure techniques. This new research area goes under the name of *Learned Data Structures*, and it was initiated in 2018 by Kraska et al. [2]. In particular, in this paper, Learned Data Structures were used mainly for the case of searching in sorted sets. This particular problem can be solved in classic algorithms by using a well-known and optimal routine, i.e., Binary Search [3,4], or more sophisticated data structures, e.g., classic indexes, such as B-Trees [5]. Usually, the classic approaches consider all of the element positions in a sorted list as possible candidates to be answers to a search query.

Such an initial list is then refined in at most $O(\log n)$ iterations, where $n$ is the size of the sorted set. The main novelty in the Learned Data Structures paradigm is the use of a machine-learning model trained over the elements of a sorted set that can learn the dataset distribution. This model uses such knowledge to make a prediction of the position of the query element in the sorted set. The prediction may be inaccurate, and thus the model returns an interval to search that accounts for prediction errors.

As a consequence, the output of the model is an interval of positions to search. The better the model, the smaller the interval. The final search stage on the reduced table positions interval is performed via Binary Search. This particular kind of Learned Data Structure is referred to as Learned Index and is the main object of this research. In what follows, in order to place our contributions in the proper context, we provide a brief literature review, followed by a presentation of our contributions to this novel area, together with a road-map of the paper in which we also highlight where it has been expanded with respect to [6].

### 1.1. Literature Review

Although the *Learned Data Structures* research field is a very novel one, it has already been extensively studied in the literature [7–9]. In what follows, we mention the main methods that can be useful for a better comprehension of the contributions provided in this paper. To this end, the most significant Learned Indexes are presented, with specific reference to their training procedures and relative benchmarking studies. Moreover, examples of real-world applications of Learned Indexes are provided, the important aspect of time/space correlation is highlighted, and, for completeness, examples of other Learned Data Structures different from Learned Indexes are given. However, the presentation is intended to be synoptic, since the interested reader can find details in the papers that are mentioned, including a recent review on the subject [7].

#### 1.1.1. Core Methods and Benchmarking Platform

The Recursive Model Index [2] (**RMI**) is the first Learned Index proposal. It is a hierarchical model that estimates the distribution of the data via a top-down approach. It can be considered as a tree-like structure, where the nodes are generic models, ranging from neural-network models [10] to simple linear or polynomial-regression models [2]. Given a query element, the internal nodes at each level identify the index of the next model (node) to use in the hierarchy. Finally, these provide a reduced interval to search into.

The tree structure of the **RMI** is characterized by the number of levels, the number of nodes for each level and the kind of models adopted at each node. As a consequence, the **RMI** depends on a consistent number of hyper-parameters, whose estimation can be a serious issue in real-world contexts as highlighted by Maltry et al. [11]. To overcome these difficulties, Marcus et al. provided a platform, referred to as **CDFShop** [12], that can be used to generate the code of a specific **RMI**, given an input dataset and specific values of its hyper-parameters. In addition, given an input dataset, the platform can provide up to ten **RMI**s.

Following the seminal proposal of the **RMI**, various new versions of Learned Indexes were designed. This is the case of the Piece-wise Geometric Model Index [13] (**PGM**) that estimates the data distribution in a bottom-up fashion by a piece-wise linear approximation algorithm [14]. Differently from the **RMI**, it is based on only one hyper-parameter $\epsilon$, which represents the maximum error admitted for the index prediction. Note that, despite the value of $\epsilon$ guaranteeing an upper bound on the search time, it does not provide any bound suggestion on the additional space used by any Learned Index with respect to the size of the input data.

The **FIT**ing-Tree model by Kraska et al. [15] was designed to overcome the mentioned space issue. It is an extension of the **PGM** using the maximum number of approximation segments as an additional parameter so that it is possible to compute the maximum additional space used by this model. Although characterized by this new space-bound

property, it is not considered in this study because of its poor performance in terms of the query time with respect to other Learned Indexes, as remarked upon in the literature [16].

The Radix Spline index [17] (**RS**) is another example of a bottom-up approach to Learned Indexing that, in a different manner to the **PGM**, estimates the distribution through a spline curve [18]. As for the **FIT**ing-Tree model, search time and space can be controlled through two hyper-parameters, i.e., the maximum error $\epsilon$ and the number of bits needed to index the spline points. However, we anticipate that such control of space is rather poor as documented by our experiments.

Except for the **PGM**, all the Learned Indexes mentioned so far are static and need to be rebuilt in the case the input dataset changes. Such a reconstruction could affect seriously the Learned Index performances, and thus a new class of indexes referred to as dynamic, was proposed. This is the case of the Adaptive Learned Index [19] (**ALEX**), which provides a Dynamic Learned Index via an extension of the **RMI**.

Due to the high number of Learned Index proposals, it is evident that it is necessary to determine the strengths and weaknesses of each method. To this end, Marcus et al. [9] provided an exhaustive benchmarking study of the main Learned Indexes on real datasets, supported by the development of a software platform referred to as *Searching on Sorted Data* [16] (**SOSD**). The mentioned study approaches the question by considering only Binary Search as the final level of Learned Indexing.

Additional pros/cons studies are available at [6,20] also considering different types of search procedures, such as Uniform Binary Search and $k$-ary Search. However, it is evident that no clear winner emerges, across the many datasets and search routines used for experimentation. It is also evident that, as also summarized in a web platform [21], in most cases, the best-performing indexes are the **RMI**, **PGM** and **RS**. As a consequence, these three Learned Indexes are the ones considered in this paper as a baseline to compare against.

### 1.1.2. Applications

Classic Indexes are widely used in a variety of real-world contexts, such as databases [22] and search engines [23]. As a consequence, Learned Indexes can also make improvements in various related applications. In particular, they are widely used for databases, providing new challenges and opportunities [24], such as the development of the so-called *Learned Databases* [25]. They have also been applied in specific kinds of databases, such as spatial [26,27] and biological [28]. Finally, another very recent application is the development of frameworks for optimizing database queries [29–32].

Once we have outlined, at a high level, the range of applications for Learned Indexes, it is appropriate to present a well-documented topic regarding the time/space trade-off of Binary Search: Static In-Memory Databases [22]. The purpose is to illustrate the importance and peculiarity of the time/space trade-off when dealing with Sorted Table Search Procedures since Static Learned Indexes speed up those routines using more space. As well-presented by Rao and Ross [22], the deployment of large databases in main memory is now possible and highly recommended since the query time greatly benefits from the data being in main memory.

Given that Binary Search has a logarithmic time performance and requires no additional space with respect to the Sorted Table, i.e., the database in main memory, Rao and Ross investigated the use of additional space in order to obtain indexing data structures that can provide faster query times with respect to Binary Search. They proposed the CSS Tree. Learned Indexes improved the query time of this data structure using less space with respect to it [13].

Analogous results hold for many other static indexing data structures. Therefore, the proper formulation of a time/space trade-off in this context is to take constant additional space with respect to the table as the space baseline and the query time of Binary Search as the time baseline. Then, as investigated by Rao and Ross, it is natural to ask if, by using more space, we can design indexes that have a query time faster than Binary Search. Learned Indexes are the most recent and effective answer to this question.

### 1.1.3. The Emergence of a New Role of Machine Learning for Data Processing and Analysis

Analogously to Learned Indexes, many methods can benefit from the combined approach of machine learning and classic data structures. An example that has been extensively discussed in the literature is the case of Bloom Filters [33], whose learned version was introduced by Kraska et al. [2], extended with several variants in [34–36] and analyzed in more depth by Fumagalli et al. [37]. Other examples are the learned versions of Hash Functions [2,38], Binary Trees [39], Rank/Select Dictionaries [40], Suffix Arrays [41] and String Dictionaries [42]. However, the importance of using a learning phase to improve the performance of a classic algorithm has not been limited only to those concerned with searching in sorted sets but has also, recently, been used for caching, scheduling and counting with data streams [8] and in the specific case of sorting operations [43].

Due to the line of research outlined thus far, a new role has emerged for machine learning in data processing and management. Indeed, it is well-established that machine-learning techniques have a broad spectrum of applications in data-driven domains as well exemplified in [44,45]. However, how to leverage those techniques to obtain improvements in the time and/or space performance of data-processing procedures that can be seen as belonging to the area of data structures and databases has been overlooked. Learned Data Structures and algorithms open the way to the exploration of machine-learning techniques to design pre-processors to boost the performance of data structures and algorithms. The implications for applications are natural and real as discussed in [46], including a better use of time and space and the ability to process larger amounts of data with fewer resources. Database systems, which are essential to any kind of large data-analysis task, can benefit the most from this new role of machine learning [47].

### 1.2. Our Contributions to Learned Indexing

As we mentioned, all Learned Index proposals offer some kind of time/space trade-off with respect to Binary Search. In turn, the abundance of those new data structures gives rise to many options regarding the use of space and time and which one to pick is specific to the application that one has in mind and the resources one has available. Unfortunately, this aspect regarding the time/space trade-off of Learned Data Structures has not been investigated in depth and rigorously following the methodology coming from classic data structures [3]. In particular, given that the intent is to speed up Binary Search with the use of additional space with respect to the input Sorted Table, we are missing an assessment of how effective constant-space models would be at speeding up Sorted Table Search procedures. In summary, two related fundamental questions have been overlooked and are stated here:

- How space-demanding should a predictive model be in order to speed up Sorted Table Search procedures.
- To what extent can one enjoy the speeding up of the search procedures provided by Learned Indexes with respect to the additional space one needs to use.

It is relevant to state that, given the previous work that we mentioned motivating the use of more space with respect to Binary Search in order to obtain better query times, the two related questions above are methodologically important since a systematic and coherent study of whether we can obtain Learned Indexes with small-space occupancy, i.e, close to constant, such as a classic Binary Search, with the characteristic of being able to speed up Sorted Table Search procedures, are not available. Moreover, such a characterization has important practical implications as discussed in [13,22].

### 1.3. Road Map of the Paper

This paper considerably expands the presentation and the results contained in [6]. In particular, we describe our research in full and provide details with an in-depth analysis. We also include material in an appendix. We provide a road map of the paper and our contributions with respect to the state of the art. Moreover, noting that the Introduction

was considerably expanded with respect to the conference version [6], we also highlight the additions that we provide in the remaining sections with regard to the conference paper [6].

Section 2 is dedicated to a formal definition of the search on sorted data problem with an outline of the classic algorithmic solution via Binary Search. Then, we provide and discuss a very simple approach to learning from data to speed up searching in sorted tables. Moreover, we propose a classification of Learned Indexes that includes two new ones as well as some that are leaders in the literature, i.e., **RMI**, **RS** and **PGM**. In particular, the first new model, referred to as *Learned k-ary Search* (**KO-US**), uses constant space, while the other new model, referred to as *Synoptic RMI* (**SY-RMI**), uses a user-defined amount of space.

The two models were introduced for the first time in [6], where they also presented an outline of their evaluation. As stated, this paper is an extended version of a conference paper [6] and provides the full spectrum of our experiments and evaluations. In regard to the conference paper [6], Section 2 here accounts for Sections 2, 3.1 and 3.2 in that paper, with very few additions in detail.

Section 3 provides our experimental methodology, which extends the one recommended in the benchmarking study by Marcus et al. [9]. In particular, in order to provide an evaluation of how Learned Indexes perform when the input table fits the different levels of the internal memory hierarchy, we extended the datasets used in the benchmarking study. This is another methodologically important contribution of this scientific research. In regard to the mentioned conference paper [6], Section 3 accounts for Section 3.4 of that paper, which is now considerably extended in terms of experimental details, and we also include material presented in Appendixes A.2 and A.4.

Section 4 describes and analyses the training phases of the two novel models. In particular, we focus on how the *Synoptic RMI* is able to learn, in small space, key features of a variety of real datasets for the purpose of prediction. Moreover, we report useful indications, overlooked so far in the literature, for Learned Index designers and practitioners about model training across different memory levels, shedding additional light on the training phase of the **RS** and the **PGM**. It is useful to recall that the **RS** was shown to be faster to train than the **PGM** only on large datasets [17]. Here, we show that, on small datasets, this is no longer the case. In regard to the conference paper [6], this section accounts for Section 4 in that paper, which is now considerably extended in terms of the description of the training phase experiments with a full discussion of the obtained results.

Section 5 describes and analyzes the Learned Indexes query phase, providing the main contributions of this paper. In particular, concerning the additional space, we analyze two possible cases: constant or nearly constant and parametric.

For the case of constant space, our main contribution is the study of the performance of the Learned *k*-ary Search Model in comparison with a Cubic Regression Model and Binary Search alone. Indeed, we anticipated that the Learned *k*-ary Search Model would perform better than the Binary Search alone and the Cubic Model, except in the case when the dataset distribution was very complex to approximate. This issue represents the main weakness of constant-space models. In addition, the Learned *k*-ary Search Model was compared with a top performing Binary-Search routine that uses a layout other than sorted, i.e., the Eytzinger Layout [23]. Our findings provide evidence that the Eytzinger Layout, when possible to use, is always competitive with respect to all the models with constant or nearly constant space—even the Learned *k*-ary Search. Unfortunately, as indicated in what follows, such a layout cannot be used within the current Learned-Indexing paradigm.

For the case of parametric space, we provide a confirmation and an extension of the findings provided in the benchmarking study by Marcus et al. [9]. Indeed, the new models introduced in the study, i.e., the Synoptic **RMI** and the bi-criteria **PGM**, perform better than the Binary Search alone, across all the datasets and memory levels, using very small additional space with respect to the input table. Moreover, even the most complex models, excluding the **RS** on the lower memory levels, achieve very good performance considering a bound of at most 10% of additional space.

We also investigate the time and space relationships of parametric models, showing that, while their query times can differ by constant factors, the corresponding spaces can disagree by several orders of magnitude. The main finding is that space seems to be the real key to the efficiency of a model. This provides additional insight into the time/space relationship of Learned Indexes, with respect to what is known in the literature. Our analysis also provides useful guidelines to practitioners interested in using Learned Indexes. The software and datasets used for this research are available at [48].

In regard to the conference paper [6], Section 5 accounts for Sections 5 and 6 of the mentioned paper, which now presents the full set of experiments on all datasets together with an in-depth analysis of the results. Some of the material relevant to this section is in Appendix A.5.

## 2. Learning from a Static Sorted Set to Speed Up Searching

Consider a sorted table $A$ of $n$ keys, taken from a universe $U$. It is well-known that Sorted Table Search can be phrased as the Predecessor Search Problem: for a given query element $x$, return the $A[j]$ such that $A[j] \leq x < A[j+1]$. With reference to such a problem, in the following, we describe the classic solutions in the literature and how to transform them into learning-prediction ones.

### 2.1. Solution with a Sorted Search Routine

It is well-known in algorithmics [1,3,4,49] that the Predecessor Search Problem can be solved with Sorted Table Search routines, such as Binary and Interpolation Search. For the aim of this paper and according to the benchmarking study, we use the C++ **lower_bound** routine, denoted as **BS** and informally referred to as Standard. In addition to this method, we use the best routines that came out of the study by Khuong and Morin [23], i.e., Uniform Binary Search [3], denoted as **US**, and Eytzinger Layout Search, denoted as **EB**.

For the convenience of the reader, details about all the above-mentioned search procedures are given in Appendix A.1. We anticipate that other routines could be considered in this study, such as Interpolation Search or its variant TIP [50]; however, the extensive experiments conducted in [51] show that they are not competitive in the Learned Indexing scenario. Therefore, in order to keep this paper focused on relevant contributions, they are omitted here.

### 2.2. Learning from Data to Speed Up Sorted Table Search: A Simple View with an Example

Kraska et al. [2] proposed an approach that transforms the Predecessor Search problem into a learning-prediction one. With reference to Figure 1, the model learned from the data is used as a predictor of where a query element may be in the table. To fix ideas, Binary Search is then performed only on the interval returned by the model.

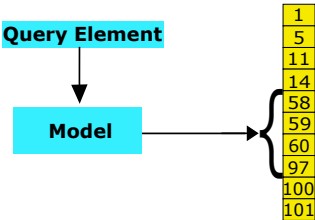

**Figure 1. A general paradigm of Learned Searching in a Sorted Table** [9]. The model is trained on the data in the table. Then, given a query element, it is used to predict the interval in the table of where to search (included in brackets in the figure).

We now outline the simplest technique that can be used to build a model for $A$ and provide an example. It relies on Linear Regression, with Mean Square Error Minimization [52]. We start with the example. Consider Figure 2 and the table $A$ in the caption.

- **Ingredient One of Learned Indexing: The Cumulative Distribution Function of a Sorted Table**. With reference to Figure 2a, we can plot the elements of *A* in a graph, where the abscissa reports the value of the elements in the table and the ordinates are their corresponding ranks. The result of the plot is reminiscent of a discrete Cumulative Distribution Function that underlines the table. The specific construction exemplified here can be generalized to any sorted table as discussed in Marcus et al. [9]. In the literature, for a given table, such a discrete curve is referenced as *CDF*.

- **Ingredient Two of Learned Indexing: A Model for the** *CDF*. Now, it is essential to transform the discrete *CDF* into a continuous curve. The simplest way to do this is to fit a straight line of equation $F(x) = ax + b$ to the *CDF* (this process is shown in Figure 2b). In this example, we use Linear Regression with Mean Square Error Minimization in order to obtain *a* and *b*. They are 0.01 and 0.85, respectively.

- **Ingredient Three of Learned Indexing: The Model Error Correction**. Since *F* is an approximation of the ranks of the elements in the table, in applying it to an element in order to predict its rank, we may produce an error *e*. With reference to Figure 2c, applying the model to the element 398, we obtain a predicted rank of 4.68, instead of 7, which is the real rank. Thus, the error made by the model $F(x) = 0.01 \times x + 0.85$ on this element is $e = 7 - \lceil 4.68 \rceil = 2$. Therefore, in order to use the equation *F* to predict where an element *x* is in the table, we must correct for this error. Indeed, we consider the maximum error $\epsilon$ computed as the maximum distance between the real rank of the elements in the table and the corresponding rank predicted by the model. The maximum error $\epsilon$ is used to set the search interval of an element *x* to be $[F(x) - \epsilon, F(x) + \epsilon]$. In the example we are discussing, $\epsilon$ is 3.

More in general, in order to perform a query, the model is consulted, and an interval in which to search is returned. Then, Binary Search is performed on that interval. Different models may use different schemes to determine the required range as outlined in Section 2.3. The reader interested in a rigorous presentation of those ideas can consult Marcus et al. [12]. In this paper, we characterize the accuracy in the prediction of a model via the *reduction factor*: the percentage of the table that is no longer considered for searching after the prediction of a rank.

Regarding the diversity across models to determine the search interval and in order to place all models on par, we empirically estimate the reduction factor of a model. With the use of the model and over a batch of queries, we determine the length of the interval to search into for each query. Based on this, we can immediately compute the reduction factor for that query. Then, we take the average of those reduction factors over the entire set of queries as the reduction factor of the model for the given table.

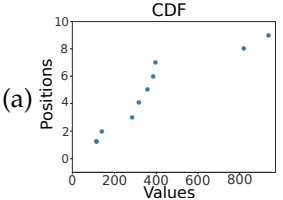 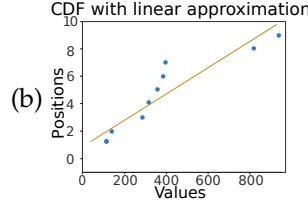 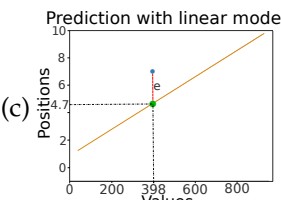

**Figure 2. The Process of Learning a Simple Model via Linear Regression.** Let table A be $[47, 105, 140, 289, 316, 358, 386, 398, 819, 939]$. (**a**) the empirical *CDF* of *A*; (**b**) the line (in orange) associated with a linear model obtained via Linear Regression; and (**c**) the error *e* made by the model in predicting the query element 398.

### 2.3. A Classification of Learned Indexing Models

With the exception of the Eytzinger Binary Search, all procedures mentioned in Section 2.1 have a natural learned version. Indeed, all models currently known in the literature naturally fit sorted table layouts for the final search stage; however, for that purpose, array layouts other than sorted or more complex data structures cannot be used. Given a learned version of the two mentioned procedures, the time and space performances depend critically on the model

used to predict the interval to search into. Here, we propose a classification of models that comprises four classes.

The first two classes, shown in Figure 3, consist of models that use constant space, while the other two, shown in Figure 4, consist of models that use space as a function of some model parameters. For each of them, the reduction factor is determined as described in Section 2. Moreover, as already indicated, the Learned *k*-ary Search and the Synoptic **RMI** models are new and fit quite naturally in the classification that we present.

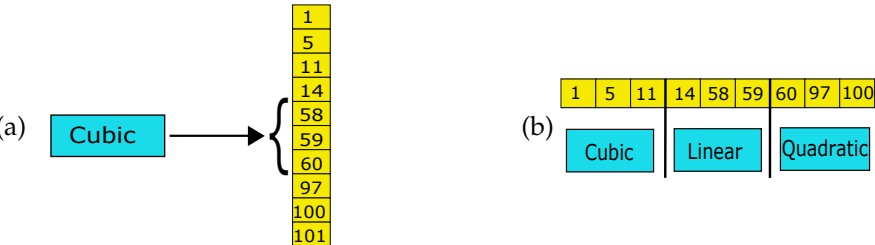

**Figure 3. Examples of various Learned Indexes that use constant space**. (**a**) An Atomic Model, where the box cubic means that the *CDF* of the entire dataset is estimated by a cubic function via regression, in analogy with the linear approximation exemplified in Figure 2. (**b**) An example of a **KO-US**, with $k = 3$. The top part divides the table into three segments, and it is used to determine the model to pick at the second stage. Each box indicates which atomic model is used for prediction on the relevant portion of the table.

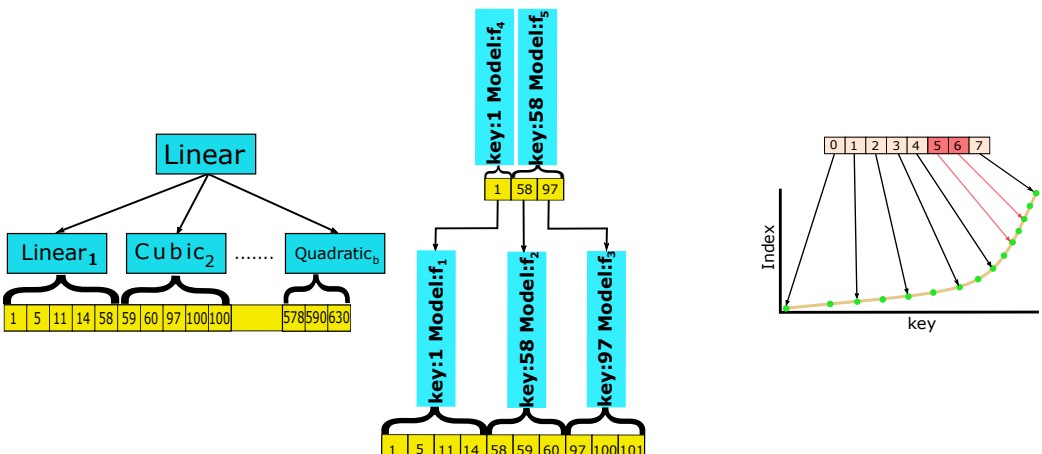

**Figure 4. Examples of various Learned Indexes that use space in functions of parameters** (see also [9]). (**Left**) An example of an **RMI** with two layers and branching factor equal to *b*. The top box indicates that the lower models are selected via a linear function. As for the leaf boxes, each indicates which Atomic Model is used for prediction on the relevant portion of the table. (**Center**) An example of a **PGM** Index. At the bottom, the table is divided into three parts. A new table is thus constructed, and the process is iterated. (**Right**) An example of an **RS** Index. At the top are the buckets where elements fall based on their three most significant digits. At the bottom, a linear spline approximating the *CDF* of the data is shown, including suitably chosen spline points. Each bucket points to a spline point so that, if a query element falls in a bucket (say six), the search interval is limited by the spline points pointed to by that bucket and the one preceding it (five in our case).

### 2.3.1. Atomic Models: One Level and No Branching Factor

- **Simple Regression** [52]. We use linear, quadratic and cubic regression models. Each can be thought of as an Atomic Model in the sense that it cannot be divided into "sub-models". Figure 3a provides an example. We report that the most appropriate regression model in terms of the query times and a reduction factor is the cubic

one. We omit those results for brevity and to keep our contribution focused on the important findings. However, they can be found in [51]. For this reason, the cubic model, indicated in the rest of the manuscript by **C**, is the only one that is included in what follows.

2.3.2. A Two-Level Hybrid Model with a Constant Branching Factor

- **KO-US: Learned $k$-ary Search**. This model partitions the table into a fixed number of segments, bounded by a small constant, i.e., at most 20 in this study, in analogy with a single iteration of the $k$-ary Search routine [53,54]. An example is provided in Figure 3b. For each segment, Atomic Models are computed to approximate the *CDF* of the table elements in that segment. Finally, the model that guarantees the best reduction factor is assigned to each segment. As for the prediction, a sequential search is performed for the second level segment to pick and the corresponding model is used for the prediction, followed by Uniform Binary Search, since it is superior to the Standard one (data not reported and available upon request). We anticipate that, for the experiments conducted in this study, $k$ is chosen in the interval $[3, 20]$. For conciseness, only results for the model with $k = 15$ are reported, since it is the value with the best performance in terms of query time (data not reported and available upon request). Accordingly, from now on, **KO-US** indicates the model with $k = 15$.

2.3.3. Two-Level RMIs with Parametric Branching Factor

- **Heuristically Optimized RMIs.** Informally, an **RMI** is a multi-level, directed graph, with Atomic Models at its nodes. When searching for a given key and starting with the first level, a prediction at each level identifies the model of the next level to use for the next prediction. This process continues until a final level model is reached. This latter is used to predict the table interval to search into. As indicated in the benchmarking study, in most applications, a generic **RMI** with two layers, a tree-like structure and a branching factor $b$ suffices. An example is provided in Figure 4 on the left. It is to be noted that the Atomic Models are **RMI**s. Moreover, the difference between Learned $k$-ary Search and **RMI**s is that the first level in the former partitions the table, while that same level in the latter partitions the universe of the elements.
  Following the benchmarking study and for a given table, we use two-layer **RMI**s that we obtain using the optimization software provided in **CDFShop**, which returns up to ten versions of the generic **RMI** for a given input table. For each model, the optimization software picks an appropriate branching factor and the type of regression to use within each part of the model—the latter quantities being the parameters that control the precision of its prediction as well as its space occupancy. It is also to be remarked, as indicated in [12], that the optimization process provides only approximations to the real optimum and is heuristic in nature with no theoretic approximation performance guarantees. The problem of finding an optimal model in polynomial time is open.
- **SY-RMI: A Synoptic RMI.** For a given set of tables of approximately the same size, we use **CDFShop** as above to obtain a set of models (at most 10 for each table). For the entire set of models thus obtained and each model in it, we compute the ratio (branching factor)/(model space), and we take the median of those ratios as a measure of the branching factor *per unit* of model space, denoted as *UB*. Among the **RMI**s returned by **CDFShop**, we pick the relative majority winner, i.e., the one that provides the best query time, averaged over a set of simulations. When one uses such a model on tables of approximately the same size as the ones used as input to **CDFShop**, we set the branching factor to be a multiple of *UB*, which depends on how much space the model is expected to use relative to the input table size. This model can be intuitively considered as the one that best summarizes the output of **CDFShop** in terms of the query time for the given set of tables. The final model is informally referred to as Synoptic.

2.3.4. CDF Approximation-Controlled Models

- **PGM** [13]. This is also a multi-stage model, built bottom-up and queried top-down. It uses a user-defined approximation parameter $\epsilon$, which controls the prediction error at each stage. With reference to Figure 4 in the center, the table is subdivided into three pieces. A prediction in each piece can be provided via a linear model guaranteeing an error of $\epsilon$. A new table is formed by selecting the minimum values in each of the three pieces. This new table is possibly again partitioned into pieces, in which a linear model can make a prediction within the given error.

  The process is iterated until only one linear model suffices, as in the case in the figure. A query is processed via a series of predictions, starting at the root of the tree. Furthermore, in this case, for a given table, at most ten models were built as prescribed in the benchmarking study with the use of the parameters, software and methods provided there, i.e, **SOSD**. It is to be noted that the **PGM** index, in its bi-criteria version, is able to return the best query time index, within the given amount of space that the model is supposed to use. Experiments are also performed with this version of the **PGM**, denoted for brevity as **B-PGM**. The interested reader can find a discussion regarding more variants of this **PGM** version in [51].

- **RS** [17]. This is a two-stage model. It also uses a user-defined approximation parameter $\epsilon$. With reference to Figure 4 on the right, a spline curve approximating the *CDF* of the data is built. Then, the radix table is used to identify spline points to use to refine the search interval. Furthermore, in this case, we performed the training as described in the benchmarking study.

In what follows, for ease of reference, models in the first two classes are referred to as constant-space models, while the ones in the remaining classes are parametric-space models.

## 3. Experimental Methodology

Our experimental setup closely follows the one outlined in the already mentioned benchmarking study by Marcus et al. [9]. Since an intent of this study is to gain deeper insights regarding the circumstances in which learned versions of Sorted Table Search procedure and indexes are profitable in small additional space with respect to the one taken by the input table, across the main memory hierarchy, we derive our own benchmark datasets from the ones in the study by Marcus et al. [9].

### 3.1. Hardware

All the experiments were performed on a workstation equipped with an Intel Core i7-8700 3.2GHz CPU with three levels of cache memory: (a) 64 kb of **L1** cache; (b) 256 kb of **L2** cache; and (c) 12 Mb of shared **L3** cache. The total amount of system memory is 32 Gbytes of DDR4. The operating system is Ubuntu LTS 20.04.

### 3.2. Datasets

The same real datasets of the benchmarking study are used. In particular, attention is restricted to integers only, each represented with 64 bits unless otherwise specified. For the convenience of the reader, a list of those datasets, with an outline of their content, is provided next.

- **amzn**: book popularity data from Amazon. Each key represents the popularity of a particular book. Although two versions of this dataset, i.e., 32-bit and 64-bit, are used in the benchmarking, no particular differences were observed in the results of our experiments, and, for this reason, we report only those for the 64-bit dataset. The interested reader can find the results for the 32-bit version in [51].
- **face**: randomly sampled Facebook user IDs. Each key uniquely identifies a user.
- **osm**: cell IDs from Open Street Map. Each key represents an embedded location.
- **wiki**: timestamps of edits from Wikipedia. Each key represents the time an edit was committed.

Moreover, for the purpose of this research, as already mentioned above, additional datasets were extracted from the ones just mentioned. For each of those datasets, three new ones were obtained in order to fit each lower level of the internal memory hierarchy. In particular, each new dataset was obtained by sampling the original one so that the *CDF* was similar to the original one. The interested reader can find more details of this extraction procedure in Appendix A.2. Letting $n$ be the number of elements in a table, for the computer architecture that was used, the details of the generated tables are the following.

- **Fitting in L1 cache: cache size 64 Kb.** Therefore, $n = 3.7K$ was chosen.
- **Fitting in L2 cache: cache size 256 Kb.** Therefore, $n = 31.5K$ was chosen.
- **Fitting in L3 cache: cache size 8 Mb.** Therefore, $n = 750K$ was chosen.
- **Fitting in PC main memory (L4): memory size 32 Gb.** Therefore, $n = 200M$ was chosen, i.e., the entire dataset.

The rationale for the choice of those datasets, in particular the ones coming from the benchmarking study, is that they provide different Empirical *CDFs*, as shown in Figure 5a, and this allows us to measure the performance of Learned Indexes considering different possible characteristics of real-world data. It is to be noted that the **face** dataset is somewhat special. Indeed, the shape of its *CDF* (see Figure 5a) is determined by 21 outliers at the end of the table: all the elements of that dataset, up to the first outlier, have essentially the same distance between consecutive elements—that is, they are all in a straight line.

This regularity breaks with the first outlier that, together with the other ones, does not follow such a nice pattern. For lower memory levels, the *CDF* of the corresponding **face** datasets becomes a straight line, as exemplified in Figure 5b for the **L3** memory level. As for the remaining datasets, their smaller versions closely follow the *CDF* of the largest datasets as again exemplified in Figure 5b for the **L3** memory level.

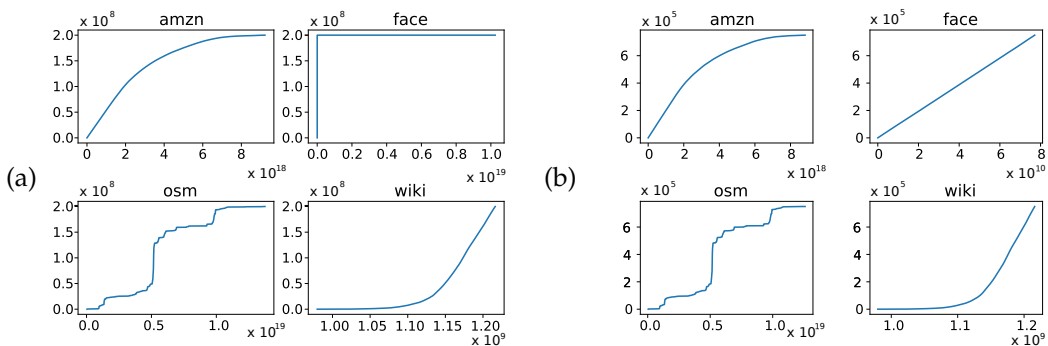

**Figure 5. The *CDF* of the main datasets**. For each dataset coming from the benchmarking study, the value of each of its elements is reported on the abscissa and its position on the ordinate. In particular, Figure (**a**) is referred to the **L4** memory level, while (**b**) to **L3**.

As for query dataset generation, for each of the tables built as described above, we extract uniformly and at random (with replacement) from the universe $U$, a total of two million elements, 50% of which are present and 50% absent, in each table. For coherence, all of the query experiments were performed within **SOSD**, suitably modified to accommodate all Learned Indexes used in this research. The query time that we report is an average taken on a batch of two million queries executed by a search routine or a Learned Index.

This is essential for Learned Indexes: a measure of a single query performance would be unreliable [55], and in fact **SOSD** does not allow it, while the method we chose is compliant with the literature [9]. Such a limitation makes it unreliable to measure certain relevant performance parameters of a Learned Index, such as, for each query, the amount of time spent for prediction and the amount of time spent for searching.

In fact, to the best of our knowledge, none of the papers reporting on Learned Indexing provided such a breakdown. Rather they concentrated on the accuracy of the predictions. For completeness, we mention that the query time estimates adopted in the current state of

the art and followed here, are accurate—that is, the processing time of a batch of queries is subject to very little standard deviation when averaged over independent executions. Although not perceived as essential in previous work, we provide a highlight of such an assessment using the new and the **RMI**, **PGM** and **RS** models on the **amzn** datasets. The results are reported in Table A8 of Appendix A.4. Finally, in terms of theoretic worst-case analysis, the prediction for the **RMI**s used here takes $O(1)$ time and $O(log n)$ time for the **PGM** and the **RS**.

## 4. Training of the Novel Models: Analysis and Insights into Model Training

We now focus on the training phase of the novel models, and we compare their performance with the literature standards included in this research. In order to assess how well a Learned Index model can be trained, three indicators are important: the time required for learning, the reduction factor that one obtains and the time needed to perform the prediction. A quantification of the first parameter is provided and discussed here. The other two indicators are strongly dependent on each other, with the reduction factor being related to space. In turn, those two indicators affect the query time. Therefore, they are best discussed in Section 5. We anticipate that our analysis of the training time performed here provides useful and novel insights into model training for Learned Indexing. All the training experiments were performed on the datasets mentioned in Section 3.2 across all internal memory levels.

### 4.1. Mining SOSD Output for the Synoptic RMI

As anticipated in Section 2.3, in order to set the levels and $UB$ of the Synoptic **RMI**, it is necessary to process the output of **SOSD** for each dataset and memory level. Indeed, as described in Section 2.3, once we set a space budget for the model, the corresponding branching factor was computed by multiplying it by $UB$. In particular, we computed three versions of a Synoptic **RMI** using a percentage of space of 0.05%, 0.7% and 2% with respect to the input table size.

With regard to the layers choice, the simulation to identify the relative majority **RMI**s was performed on query datasets extracted as described in the previous Section but using only 1% of the number of query elements specified there. The statistics regarding the results of such a simulation are summarized in Figure 6. In particular, for each memory level, we report the computed $UB$. Furthermore, limited to the top layer of an **RMI**, we also report the models associated with the best ones. The time it took to identify the proper Synoptic **RMI** (average time per element, over all the **RMI**s returned by **CDFShop**, denoted as the mining time) is also reported, together with the same time required to obtain the output of **CDFShop**.

As is evident, the mining time is comparable with the performance of **CDFShop**. It is also evident from that figure that the variety of best-performing models represents various challenges for the learning of the *CDF* of real datasets. Therefore, given such a variety, it is far from obvious that the median $UB$ is the same for each memory level. Moreover, the relative majority model is also the same across memory levels, i.e., linear spline, with linear models for each segment of the second layer.

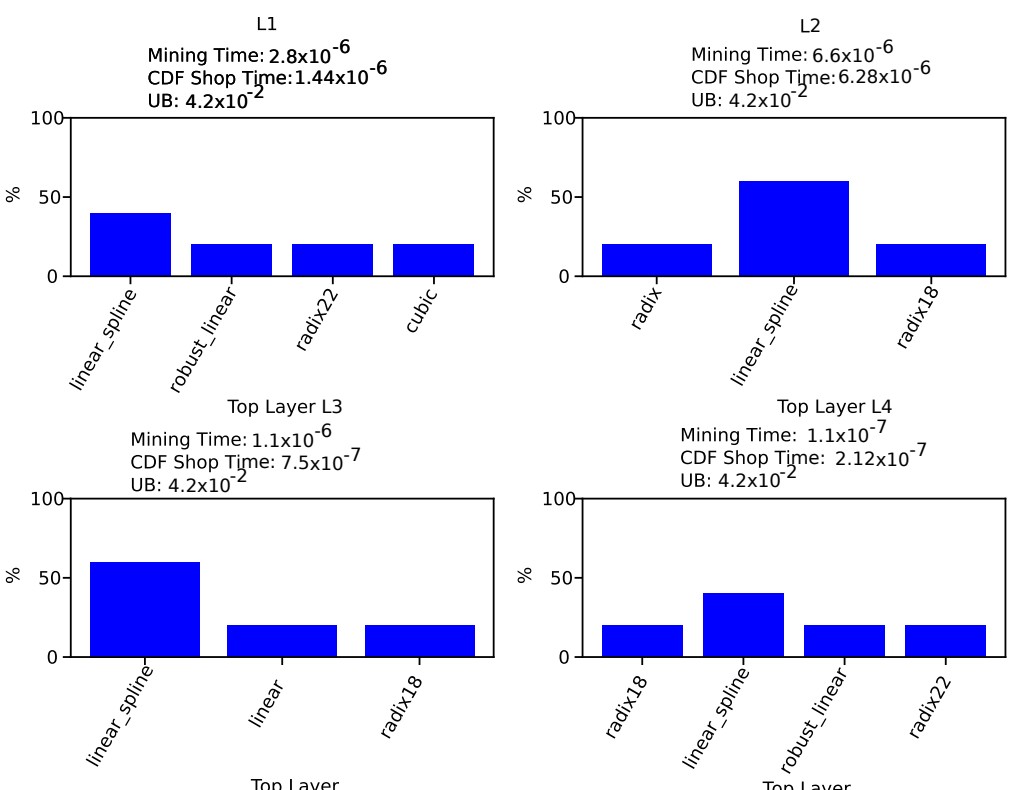

**Figure 6. Time and *UB* for the identification of the Synoptic RMIs.** For each memory level, only the top layer of the various models is indicated in the abscissa, while the ordinate indicates the number of times, in percentage, that the given model is the best in terms of the query performance on a table. On top of each histogram, we report the branching factor per unit of space as well as the mining time (in seconds) to build the synoptic models. For comparison, we also report the same time spent in obtaining the output of **CDFShop**.

*4.2. Training Time Comparison Between Novel Models and the State of the Art*

In what follows, we divide the training-time comparison into two groups: constant- and parametric-space models. For the first group, we consider the new model and only the Cubic Atomic Model, excluding the linear and quadratic ones for the reasons mentioned earlier in this paper. For the Cubic Model, the training time on a given dataset is due to the computation of its parameters via polynomial regression. As for the Learned *k*-ary Search Model, its training consists of partitioning the table into *k* segments. Then, for each segment, Atomic Models are used to approximate the local *CDF* of the elements belonging to that segment, and, among them, the model with the best reduction factor is chosen.

For each dataset and each memory level, the resulting training times are reported in Table 1 and Tables A2–A4 of Appendix A.3. As expected, the Learned *k*-ary Search Model is slower than the Cubic Atomic Model; however, the important fact is that the slowdown is due to constant multiplicative factors rather than being of an order of magnitude—that is, the slow-down is tolerable. Another additional and counter-intuitive finding is that the training time of both models, on average, is better for the cases of large datasets with respect to smaller ones. We analyzed the training code in order to obtain insight into such a fact. We found that the cost of the matrix products involved in the training computation of both models depends on the size of the involved operands. As the size of the dataset grows, such a cost is amortized on a larger and larger number of elements.

**Table 1. Constant-space model training time for the L4 tables**. The first column indicates the datasets. The remaining columns indicate the model used for the learning phase. Each entry reports the training time in seconds and per element.

|  | KO-US | C |
|---|---|---|
| **amzn** | $3.7 \times 10^{-8}$ | $1.4 \times 10^{-8}$ |
| **face** | $3.6 \times 10^{-8}$ | $1.4 \times 10^{-8}$ |
| **osm** | $3.6 \times 10^{-8}$ | $1.4 \times 10^{-8}$ |
| **wiki** | $3.6 \times 10^{-8}$ | $1.4 \times 10^{-8}$ |

Regarding the second group, we consider the new model and the ones described in Section 2.3, i.e., **RMI**, **PGM** and **RS**. The training time was computed using two different platforms: **CDFShop** in the case of the Synoptic **RMI** and **RMI** as well as **SOSD** for **PGM** and **RS**. It is useful to recall that, in the case of the state-of-the-art models, the result of a single execution of those two platforms returns a batch of up to ten models, and thus the reported times refer to the execution of the entire learning suite—that is, the training of those models consists of a batch of model instances from which a user can choose.

On the other hand, for the Synoptic **RMI**, this refers to the training of a single **RMI** with a given branching factor and layer composition. For each dataset and each memory level, the results are reported in Table 2 and Tables A5–A7 of Appendix A.3. The time needed to train the Synoptic **RMI** is comparable to the one needed to train a batch of **RMI** models. This latter, as already known, is worse than the time to train a batch of **RS** or **PGM** models. Such results are not considered problematic for the deployment of the **RMI** in application contexts, and the training time of the Synoptic **RMI** is in-line with the mentioned literature standards. For completeness, we mention that the reason for which the time needed to train a unique Synoptic **RMI** model is very close to the training of a batch on **RMI** models is due to the library start-up overhead time.

Such a time is mitigated for the case of the training of a batch of models, while it becomes dominant in training a single model. Fortunately, the **CDFShop** or the **SOSD** training executions are a "one-time-only" processes, in which the output can then be reused over and over again, suggesting that this overhead time is of little relevance for the case of a production environment.

**Table 2. Parametric-model training time for the L4 tables**. The first column indicates the datasets. The remaining columns indicate the model used for the learning phase. In particular, each entry reports the time to train the Synoptic **RMI** and an entire batch of models via the **CDFShop** and the **SOSD** libraries as specified in the main text. The time is in seconds and per element.

|  | CDFShop SY-RMI 2% | CDFShop RMI | SOSD RS | SOSD PGM |
|---|---|---|---|---|
| **amzn** | $1.1 \times 10^{-6}$ | $2.2 \times 10^{-6}$ | $2.1 \times 10^{-7}$ | $5.0 \times 10^{-7}$ |
| **face** | $1.3 \times 10^{-6}$ | $2.5 \times 10^{-6}$ | $2.1 \times 10^{-7}$ | $6.5 \times 10^{-7}$ |
| **osm** | $1.2 \times 10^{-6}$ | $2.5 \times 10^{-6}$ | $2.2 \times 10^{-7}$ | $7.4 \times 10^{-6}$ |
| **wiki** | $1.1 \times 10^{-6}$ | $2.2 \times 10^{-6}$ | $1.9 \times 10^{-7}$ | $4.1 \times 10^{-7}$ |

*4.3. Insights into the Training Time of the **RS** and **PGM** Models*

Another important contribution that this research provides is a more refined assessment of the relation between the **RS** and **PGM** indexes, in terms of the training time. In Table 3, for each dataset and memory level, we report the training times of those two indexes. As discussed in [17], those two Learned Indexes can both be built in one pass over the input with important implications: one being that they can be trained faster than the **RMI**s—even one order of magnitude sped up. However, in that study as well as

in the benchmarking one, the **RS** was reported as superior to the **PGM** in terms of the training time.

It is to be noted that the datasets that they used are the largest ones in this study. With reference to Table 3, our experiments confirm such a finding. On the other hand, the **PGM** is more effective in terms of the training time across the lower memory hierarchy. The reason may be the following. Those two indexes both use streaming procedures in order to approximate the *CDF* of the input dataset within a parameter $\epsilon$ via the use of straight-line segments that partition the universe. The main difference between the two is that the latter finds an optimal partition, determined via a well-known algorithm (see references in [13]), while the former finds a partition that approximates the optimal as described in [18]. Such an approximation algorithm is supposed to be faster than the optimal one; however, apparently, this speed pays off on large datasets.

**Table 3. Comparison between the RS and PGM training times**. For each dataset and memory level, we report the training times for the **RS** and **PGM** models in seconds. Panel (a) refers to memory levels **L1** and **L2**, while Panel (b) refers to **L3** and **L4**.

| | Panel (a) | | | |
| --- | --- | --- | --- | --- |
| | L1 | | L2 | |
| | **SOSD RS** | **SOSD PGM** | **SOSD RS** | **SOSD PGM** |
| **amzn** | $3.5 \times 10^{-6}$ | $5.0 \times 10^{-7}$ | $3.5 \times 10^{-7}$ | $5.0 \times 10^{-8}$ |
| **face** | $1.1 \times 10^{-6}$ | $3.9 \times 10^{-7}$ | $1.1 \times 10^{-7}$ | $3.9 \times 10^{-8}$ |
| **osm** | $6.9 \times 10^{-6}$ | $4.0 \times 10^{-7}$ | $6.9 \times 10^{-7}$ | $4.0 \times 10^{-8}$ |
| **wiki** | $1.0 \times 10^{-5}$ | $3.7 \times 10^{-7}$ | $1.0 \times 10^{-6}$ | $3.7 \times 10^{-8}$ |
| | Panel (b) | | | |
| | L3 | | L4 | |
| | **SOSD RS** | **SOSD PGM** | **SOSD RS** | **SOSD PGM** |
| **amzn** | $2.4 \times 10^{-8}$ | $3.4 \times 10^{-8}$ | $2.1 \times 10^{-7}$ | $5.0 \times 10^{-7}$ |
| **face** | $1.4 \times 10^{-8}$ | $2.4 \times 10^{-8}$ | $2.1 \times 10^{-7}$ | $6.5 \times 10^{-7}$ |
| **osm** | $3.5 \times 10^{-8}$ | $3.8 \times 10^{-8}$ | $2.2 \times 10^{-7}$ | $7.4 \times 10^{-7}$ |
| **wiki** | $5.1 \times 10^{-8}$ | $3.7 \times 10^{-8}$ | $1.9 \times 10^{-7}$ | $4.1 \times 10^{-7}$ |

## 5. Query Experiments

The query experiments were performed using all the methods described in Sections 2.1 and 2.3. The query datasets were generated as described in Section 3.2, and the models were trained as described in Section 4. Following that section, we divided the presentation of the query experiments and the relative discussion into two groups. For both groups, for conciseness, we report here only the experiments for the **amzn** and the **osm** datasets since they are representative of two different levels of difficulty in learning their *CDF*s. The results regarding the other datasets are reported in Appendix A.5.

### 5.1. Constant-Space Models

The results of the experiments for this group of models are reported in Figures 7 and 8 for the **amzn** and the **osm** datasets, respectively, and in Figures A2 and A3 of the Appendix A.5, for the remaining ones. In those figures, only the query time for Uniform Binary Search is reported, since the results are analogous to the ones obtained by using the Standard routine. In addition, the query time for the Eytzinger Binary Search is also reported as a useful baseline due to its superiority among the classic routines that take constant additional space with respect to the size of the input table as discussed in [23]. From the mentioned figures, it is evident that the query performance of each model

considered here is highly influenced by how difficult to learn the *CDF* of the input table is as explained next.

- The Cubic Model achieved a high reduction factor, i.e., $\approx 99\%$, on the versions of the **face** dataset for the first three levels of the internal memory hierarchy, and it was also the best performing, even compared to the Eytzinger Layout routine. This is a quite remarkable achievement; however, the involved datasets had an almost uniform *CDF*, while a few outliers disrupt this uniformity on the **L4** version of that dataset (see Figure 5 and the discussion regarding the **face** dataset in Section 3.2).

- The Learned *k*-ary Search Model achieved a high reduction factor on all versions of the **amzn** and the **wiki** datasets, i.e., $\approx 99.73$ and was faster than the Uniform Binary Search and the Cubic Model. Those datasets have a regular *CDF* across all the internal memory levels. It is to be noted that the Eytzinger Layout routine is competitive with the Learned *k*-ary Search Model.

- No constant space Learned Model won on the difficult-to-learn dataset. The **osm** dataset has a difficult-to-learn *CDF* (see Figure 5), and such a characteristic is preserved across the internal memory levels. The Learned *k*-ary Search Model achieved a respectable reduction factor, i.e., $\approx 98\%$, but no speed increase with respect to Uniform Binary Search. In order to obtain insights into such a counter-intuitive behavior, we performed an additional experiment.

  For each representative dataset and as far as the Learned *k*-ary Search Model is concerned, we computed two kinds of reduction factors: the first was the global one, achieved considering the size of the entire table, while the second was the local one, computed as the average among the reduction factors of each segment. Those results are reported in Table 4. For the **osm** dataset, it is evident that the local reduction factors are consistently lower than the global ones, highlighting that its *CDF* is also locally difficult to approximate, which, in turn, implies an ineffective use of the local prediction for the Learned *k*-ary Search, resulting in poor time performance. Finally, it is to be noted that the Eytzinger Layout routine was the best performing.

In conclusion, in applications where there is a constant space constraint with respect to the input table and where a layout other than sorted can be used, then the Eytzinger Binary Search is still the best choice unless the *CDF* of the input dataset is particularly easy to approximate. If such a layout cannot be afforded, the best choice is the use of a constant-space model, in particular the Learned *k*-ary Search Model only for datasets with a *CDF* that is simple to approximate, otherwise the use of Uniform Binary Search alone is indicated.

Our research extends the results in [23] regarding the Eytzinger Binary Search routine: even compared to Learned Indexes that use constant space, it still results as competitive and many times superior to the others.

**Table 4. Global and local reduction factors**. For the two representative datasets, i.e., **amzn** and **osm**, and for each memory level, in each entry, we report the global reduction factor (left) and the local one (right).

|  | amzn | osm |
|---|---|---|
| **L1** | 99.94−99.48 | 98.12−86.70 |
| **L2** | 99.98−99.56 | 98.07−86.57 |
| **L3** | 99.98−99.53 | 97.98−86.43 |
| **L4** | 99.98−99.54 | 98.03−86.57 |

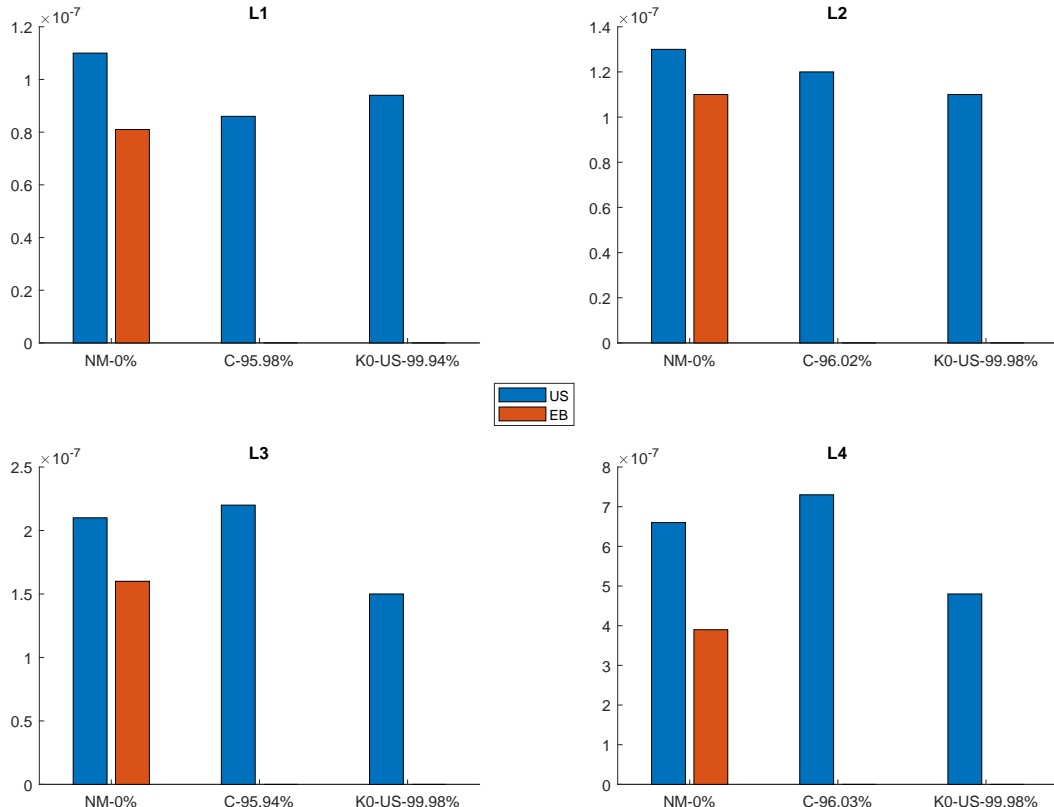

**Figure 7. Constant-space-model query times for the amzn dataset.** For each memory level, the blue bar reports the average query time in seconds of Uniform Binary Search using, from left to right, no model, Cubic model and **KO-US**. In addition, we report the average query time also for the Eytzinger Binary Search in the orange bar.

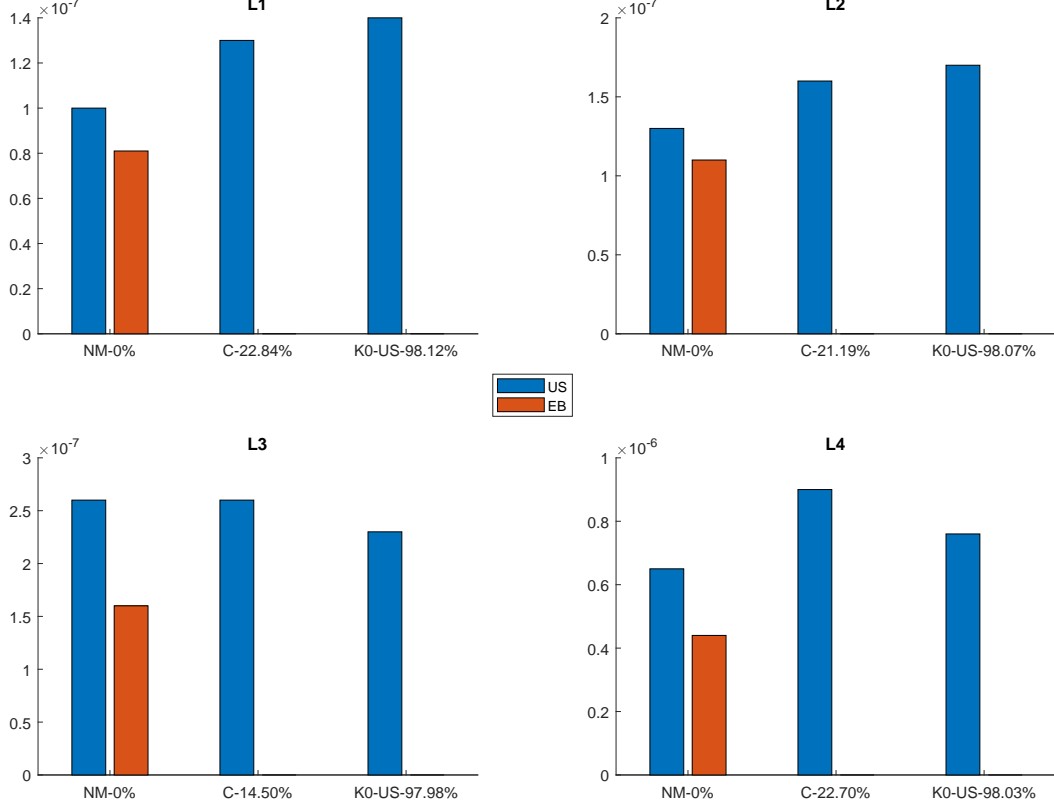

**Figure 8. Constant-space-model query times for the osm dataset.** The figure legend is as in Figure 7.

### 5.2. Parametric-Space Models

For the convenience of the reader, we recall that the model classes involved are: **RMI**, **RS**, **PGM**, the Synoptic **RMI** and the bi-criteria **PGM**, which are trained on the input datasets (see Section 3.2) as reported in Section 4. The batch of queries used here was obtained as described in Section 3.2 and, for the query time, we took the average per element as specified in that section. For each of the first three model classes, we considered, among the trained models, the fastest in terms of the query time and those taking less than 10% of space with respect to the the space taken by the input table.

For the other two model classes, we considered three increasing bounds on space, i.e., 0.05%, 0.7% and 2%, with respect to the space of the table alone. Moreover, as a measure of the Learned Indexes speed up, we also report the query time of the Uniform Binary Search. The results of the corresponding experiments are reported in Figures 9 and 10 for the **amzn** and **osm** datasets, respectively, and in Figures A4 and A5 of the Appendix A.5 for the remaining ones.

An interesting finding is that both the Synoptic **RMI** and the bi-criteria **PGM** performed better than Uniform Binary Search across datasets and memory levels using very little additional space—that is, one can enjoy the speed of Learned Indexes with a very small space penalty. Moreover, it is important to note that, except for the **L1** memory level, the space of those two models is very close to the user-defined bound. Furthermore, in terms of query performances, such two models seem to be complementary. In fact, the bi-criteria **PGM** performed better on the **L1** and **L4** memory levels, while the Synoptic **RMI** performed better on the remaining ones. This complementary and effective control of space makes these two models quite useful in practice.

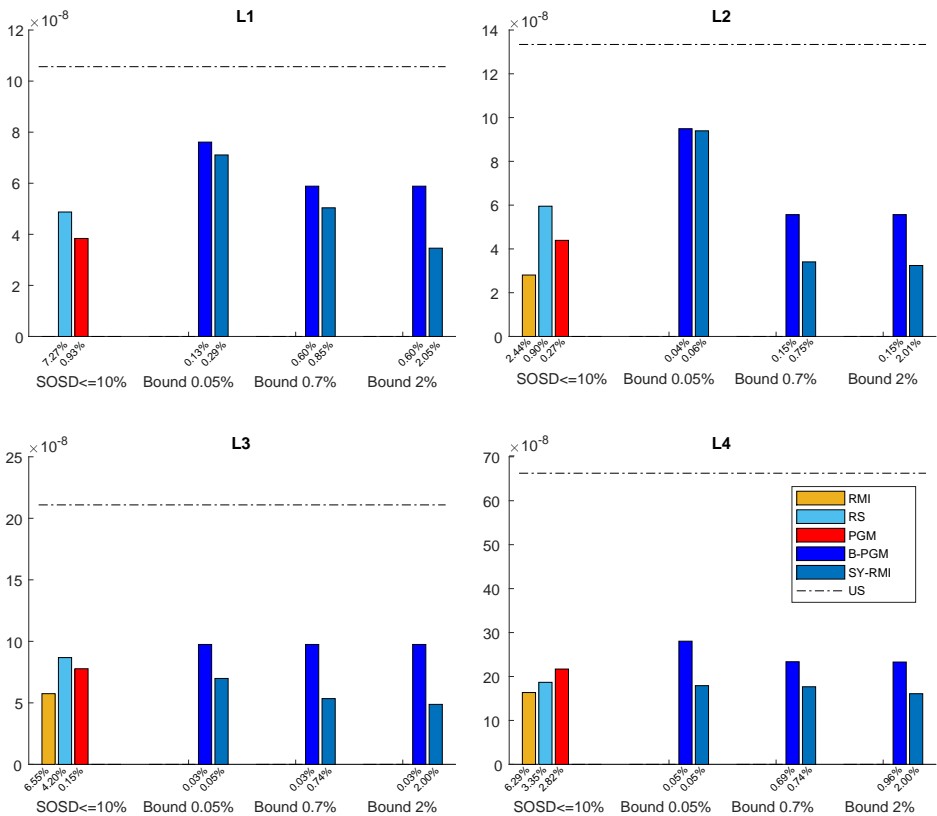

**Figure 9. Query times for the amzn dataset on Learned Indexes in small space.** The methods are the ones in the legend (the middle of the four panels; the notation is as in the main text, and each method has a distinct color). For each memory level, the abscissa reports methods grouped by space occupancy as specified in the main text. When no model in a class output by **SOSD** took at most 10% of additional space, that class was left absent. The ordinate reports the average query time in seconds, with Uniform Binary Search executed in **SOSD** as a baseline (horizontal lines).

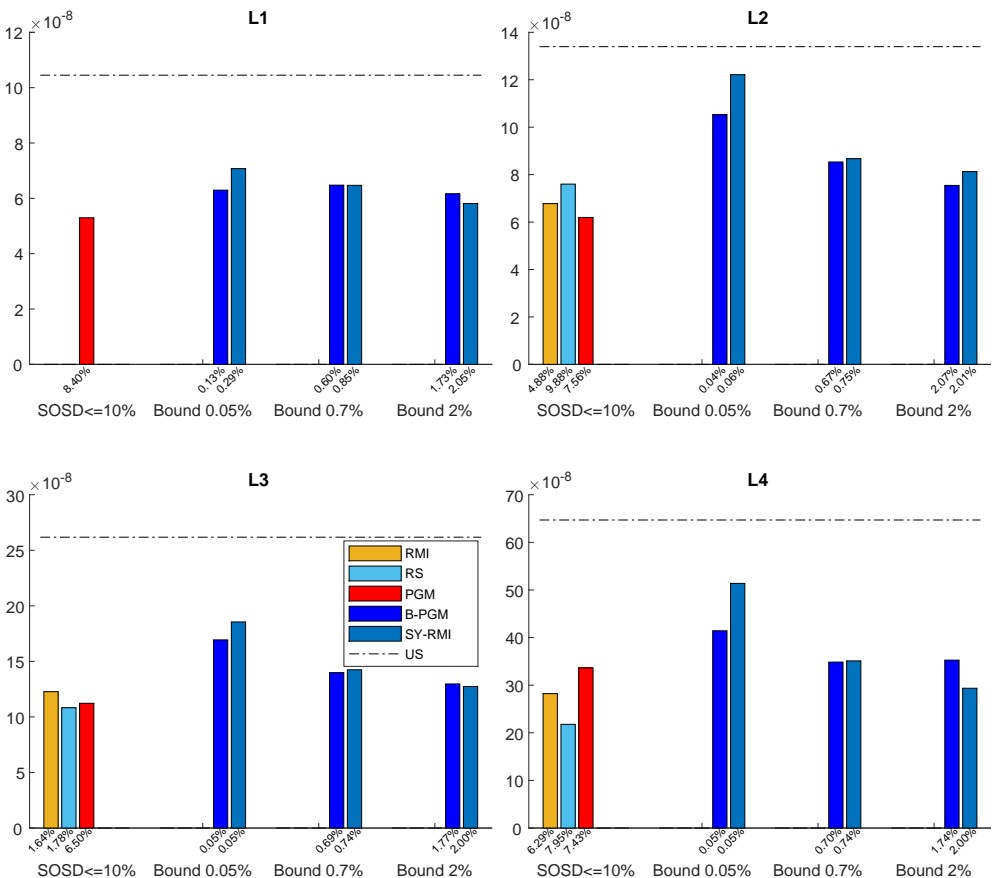

**Figure 10. Query times for the osm dataset on Learned Indexes in small space.** The figure legend is as in Figure 9.

In addition to those findings, our research provides some more insights into the time–space relation in Learned Indexes, thereby, extending the results of the benchmark study as we now discuss.

- **Space Constraints and the Models Provided by SOSD.** We fixed a rather small space budget, i.e., at most 10% of additional space in order for a model returned by **SOSD** to be considered. The **RS** Index was not competitive with respect to the other Learned Indexes. Those latter consistently use less space and time across datasets and memory levels. As for the **RMI**s coming out of **SOSD**, they were not able to operate in a small space at the **L1** memory level. On the other memory levels, they were competitive with respect to the bi-criteria **PGM** and the Synoptic **RMI**; however, they required more space with respect to them.
- **Space, Time and Accuracy of Models.** As stated in the benchmarking study, a common view of Learned Indexing Data Structures is as a *CDF* lossy compressor; see also [2,13]. In this view, the quality of a Learned Index can be judged by the size of the structure and its reduction factor. In that study, it was also argued that this view does not provide an accurate selection criterion for Learned Indexes. Indeed, it may very well be that an index structure with an excellent reduction factor takes a long time to produce a search bound, while an index structure with a worse reduction factor that quickly generates an accurate search bound may be of better use. In the benchmarking study, they also provided evidence that the space–time trade-off is the key factor in determining which model to choose.

  Our contribution is to provide additional results supporting those findings. To this end, we conducted several experiments, whose results are reported in Tables 5 and 6 and Tables A9 and A10 in the Appendix A.5. In these Tables, for each dataset, we report a synopsis of three parameters, i.e., the query time and space used in addition

by the model and reduction factor, across all datasets and memory levels. In particular, for each dataset, we compare the best-performing model with all the ones that use small space, taking, for each parameter, the ratio of the model/best model. The ratio values are reported from the second row of the table, and the first row shows the average values of the parameters for the best model.

First, it is useful to note that, even in a small-space model, it is possible to obtain a good, if not nearly perfect, prediction (i.e., a very high reduction factor). However, prediction power is somewhat marginal to assess performance. Indeed, across memory levels, we see a space classification of model configurations. The most striking feature of this classification is that the gain in query time between the best model and the others is within small constant factors, while the difference in space occupancy may be, in most cases, several orders of magnitude different—that is, space is the key to efficiency.

**Table 5. A synoptic table of space, time and accuracy of the models on the amzn dataset**. For each memory level, we report, in the first row, the best performing method for that memory level. The columns named time, space and reduction factor indicate, for this best model, the average query time in seconds, the average additional space used in Kb and the average of the empirical reduction factor. In the second row, we report the versions of the **RMI**, **RS**, **PGM** and Synoptic **RMI** models that use the least space. In particular, the number next to the models represents, in percentage, the bound on the used space with respect to the input dataset. The columns indicate the ratio of the model/best model of the time, space and reduction factor.

| | Time | Space | Reduction Factor |
|---|---|---|---|
| **L1** | | | |
| | Time | Space | Reduction Factor |
| Best **RMI** | $1.89 \times 10^{-8}$ | 3.09 | 99.84 |
| **B-PGM** 0.05 | 4.03 | $1.30 \times 10^{-2}$ | $2.50 \times 10^{-1}$ |
| **SY-RMI** 0.05 | 3.77 | $2.85 \times 10^{-2}$ | $1.75 \times 10^{-1}$ |
| **RS** $< 10$ | 2.58 | $7.06 \times 10^{-1}$ | $9.29 \times 10^{-1}$ |
| Best **RMI** | 1.00 | 1.00 | 1.00 |
| **L2** | | | |
| | Time | Space | Reduction Factor |
| Best **RMI** | $2.51 \times 10^{-8}$ | 6.16 | 99.97 |
| **B-PGM** 0.05 | 3.78 | $1.62 \times 10^{-2}$ | $9.16 \times 10^{-1}$ |
| **SY-RMI** 0.05 | 3.74 | $2.60 \times 10^{-2}$ | $6.44 \times 10^{-1}$ |
| Best **RS** $< 10$ | 2.38 | $3.68 \times 10^{-1}$ | $9.92 \times 10^{-1}$ |
| Best **RMI** | 1.00 | 1.00 | 1.00 |
| **L3** | | | |
| | Time | Space | Reduction Factor |
| Best **RMI** | $4.70 \times 10^{-8}$ | $6.29 \times 10^{3}$ | 100.00 |
| **B-PGM** 0.05 | 2.07 | $3.05 \times 10^{-4}$ | 1.00 |
| **SY-RMI** 0.05 | 1.49 | $4.79 \times 10^{-4}$ | $9.99 \times 10^{-1}$ |
| **RS** $< 10$ | 1.59 | $4.00 \times 10^{-2}$ | 1.00 |
| **RMI** $< 10$ | 1.03 | $6.25 \times 10^{-2}$ | 1.00 |

**Table 5.** *Cont.*

|  | L4 | | |
|---|---|---|---|
|  | **Time** | **Space** | **Reduction Factor** |
| Best **RMI** | $1.51 \times 10^{-7}$ | $2.01 \times 10^5$ | 100.00 |
| **B-PGM** 0.05 | 1.85 | $3.93 \times 10^{-3}$ | 1.00 |
| **SY-RMI** 0.05 | 1.18 | $3.97 \times 10^{-3}$ | 1.00 |
| Best **RS** | 1.19 | $7.16 \times 10^{-2}$ | 1.00 |
| **RMI** $< 10$ | 1.03 | $5.00 \times 10^{-1}$ | 1.00 |

**Table 6. A synoptic table of space, time and accuracy of the models on the osm dataset**. The legend is as in Table 5.

|  | L1 | | |
|---|---|---|---|
|  | **Time** | **Space** | **Reduction Factor** |
| Best **RMI** | $2.72 \times 10^{-8}$ | $1.15 \times 10^3$ | 99.87 |
| **B-PGM** 0.05 | 2.31 | $3.49 \times 10^{-5}$ | $1.74 \times 10^{-1}$ |
| **SY-RMI** 0.05 | 2.60 | $7.67 \times 10^{-5}$ | $2.30 \times 10^{-1}$ |
| Best **RMI** | 1.00 | 1.00 | 1.00 |
| Best **RS** | 1.19 | $4.33 \times 10$ | $9.99 \times 10^{-1}$ |
|  | **L2** | | |
|  | **Time** | **Space** | **Reduction Factor** |
| Best **RMI** | $3.93 \times 10^{-8}$ | $1.84 \times 10^3$ | 99.97 |
| **B-PGM** 0.05 | 2.68 | $5.45 \times 10^{-5}$ | $7.75 \times 10^{-1}$ |
| **SY-RMI** 0.05 | 3.11 | $8.72 \times 10^{-5}$ | $7.24 \times 10^{-1}$ |
| **RMI** $< 10$ | 1.73 | $6.71 \times 10^{-3}$ | $9.87 \times 10^{-1}$ |
| **RS** $< 10$ | 1.93 | $1.36 \times 10^{-2}$ | $9.79 \times 10^{-1}$ |
|  | **L3** | | |
|  | **Time** | **Space** | **Reduction Factor** |
| Best **RS** | $7.06 \times 10^{-8}$ | $4.63 \times 10^4$ | 100.00 |
| **B-PGM** 0.05 | 2.40 | $6.22 \times 10^{-5}$ | $9.98 \times 10^{-1}$ |
| **SY-RMI** 0.05 | 2.63 | $6.52 \times 10^{-5}$ | $9.31 \times 10^{-1}$ |
| **RMI** $< 10$ | 1.75 | $2.12 \times 10^{-3}$ | $9.97 \times 10^{-1}$ |
| **RS** $< 10$ | 1.55 | $2.31 \times 10^{-3}$ | 1.00 |
|  | **L4** | | |
|  | **Time** | **Space** | **Reduction Factor** |
| Best **RS** | $2.04 \times 10^{-7}$ | $5.08 \times 10^5$ | 100.00 |
| **SY-RMI** 0.05 | 2.52 | $1.57 \times 10^{-3}$ | $9.99 \times 10^{-1}$ |
| **B-PGM** 0.05 | 2.03 | $1.59 \times 10^{-3}$ | 1.00 |
| **RMI** $< 10$ | 1.18 | $1.98 \times 10^{-1}$ | 1.00 |
| **RS** $< 10$ | 1.05 | $2.50 \times 10^{-1}$ | 1.00 |

## 6. Conclusions and Future Directions

In this research, we provided a systematic experimental analysis regarding the ability of Learned Model Indexes to perform better than Binary Search in small space. This is

the first step forward in the full characterization of the time/space trade-off spectrum of Learned Indexes, with respect to Sorted Table Search routines that use constant additional space.

In particular, in regard to the first question stated in Section 1.2, i.e., how space-demanding should be a predictive model in order to speed up Sorted Table Search procedures, we show that constant-space models may grant such a speeding up with respect to classic versions of Binary Search unless the data *CDF* to be learned is complex. In addition, we also show that models using a small percentage of additional space with respect to the Sorted Table guarantee consistent speed ups of sorted layout Binary Search procedures across Learned Indexes and datasets with different levels of *CDF* complexity to learn.

In regard to the second question, i.e., to what extent one can enjoy the speed up of the search procedures provided by Learned Indexes with respect to the additional space one needs to use, our experiments bring to light the existence of a large gap between the best-performing methods and the others that we considered and that operate in small space. Indeed, the query time performance of the latter with respect to the former is bounded by small constants, while the space usage may differ even by five orders of magnitude.

These findings bring to light the acute need to investigate the existence of small-space models that should close the time gap mentioned earlier. Another important aspect, with potential practical impacts, is to devise models that can work on layouts other than Sorted, i.e., Eytzinger. Indeed, since the Eytzinger layout is consistently faster than the sorted ones [23], it would be of interest to provide models that take advantage of this layout rather than the sorted ones.

**Author Contributions:** D.A., R.G. and G.L.B. contributed equally to the design of the algorithms and the analysis of the experiments. D.A. developed most of the code and performed most of the experiments. R.G. wrote a substantial part of the paper, with substantial contributions by D.A. and G.L.B. to the presentation of the results in graphic and tabular form. All authors have read and agreed to the published version of the manuscript.

**Funding:** This research was funded in part by the MIUR Project of National Relevance 2017WR7SHH "Multicriteria Data Structures and Algorithms: from compressed to learned indexes, and beyond". Additional support to RG was granted by Project INdAM—GNCS "Modellizazzione ed analisi di big knowledge graphs per la risoluzione di problemi in ambito medico e web".

**Data Availability Statement:** All datasets are publicly available. The largest ones are accessible as stated in [9]. The datasets generated for this research are available at [56].

**Conflicts of Interest:** The authors declare no conflict of interest.

## Abbreviations

The following abbreviations are used in this manuscript:

| | |
|---|---|
| CDF | Cumulative Distribution Function |
| RMI | Recursive Model Index |
| PGM | Piece-wise Geometric Model Index |
| RS | Radix Spline index |
| ALEX | Adaptive Learned index |
| SOSD | Searching on Sorted Data |
| KO-US | Learned *k*-ary Search |
| SY-RMI | Synoptic RMI |
| BS | lower_bound search routine |
| US | Uniform Binary Search |
| EB | Eytzinger Layout Search |

| | |
|---|---|
| C | Cubic regression model |
| B-PGM | Bicriteria Piece-wise Geometric Model Index |
| L1 | cache of size 64kb |
| L2 | cache of size 256kb |
| L3 | cache of size 12Mb |
| L4 | memory size 32Gb |
| amzn | the Amazon dataset |
| face | the Facebook dataset |
| osm | the OpenStreetMap dataset |
| wiki | the Wikipedia dataset |

## Appendix A. Methods and Results: Additional Material

*Appendix A.1. Binary Search and Its Variants*

With reference to the routines mentioned in the main text (Section 2.1) and following the research in [23], we provide more details about two kind of layouts.

1. Sorted. We use two versions of Binary Search for this layout. The template of the **lower_bound** routine is provided in Algorithm A1, while the Uniform Binary Search implementation is given in Algorithm A2. In particular, this implementation of Binary Search is as found in [23].
2. Eytzinger Layout [23]. The sorted table is now seen as stored in a virtual complete balanced binary search tree. Such a tree is laid out in Breadth-First Search order in an array. An example is provided in Figure A1. The implementation is reported in Algorithm A3.

---

**Algorithm A1 lower_bound Template.**

---

1: ForwardIterator lower_bound (ForwardIterator first, ForwardIterator last, const T& val){
2:   ForwardIterator it;
3:   iterator_traits<ForwardIterator>::difference_type count, step;
4:   count = distance(first,last);
5:   while (count>0){
6:     it = first; step=count/2; advance (it,step);
7:     if (*it<val){
8:       first=++it;
9:       count-=step+1;
10:     }
11:     else count=step;
12:   }
13:   return first;
14: }

---

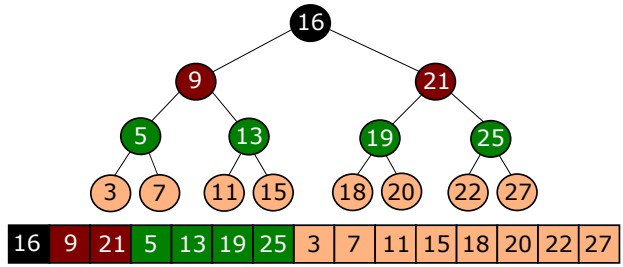

**Figure A1. An example of Eyzinger layout of a table with 15 elements.** Nodes with the same color in the tree are contiguous in the array. See also [23].

---

**Algorithm A2 Implementation of Uniform Binary Search.** The code is as in [23] (see also [3,54]).

```
 1: int UniformBinarySearch(int *A, int x, int left, int right){
 2:    const int *base = A;
 3:    int n = right;
 4:    while (n > 1) {
 5:      const int half = n / 2;
 6:      __builtin_prefetch(base + half/2, 0, 0);
 7:      __builtin_prefetch(base + half + half/2, 0, 0);
 8:      base = (base[half] < x) ? &base[half] : base;
 9:      n -= half;
10:    }
11:    return (*base < x) + base - A;
12: }
```

---

**Algorithm A3 Uniform Binary Search with Eytzinger layout.** The code is as in [23].

```
    int EytzingerLayoutSearch(int *A, int x, int left, int right){
      int i = 0;
 3:   int n = right;
      while (i < n){
        __builtin_prefetch(A+(multiplier*i + offset));
 6:     i = (x <= A[i]) ? (2*i + 1) : (2*i + 2);
      }
      int j = (i+1) >> __builtin_ffs(~(i+1));
 9:   return (j == 0) ? n : j-1;
    }
```

---

*Appendix A.2. Datasets: Details*

With reference to the datasets mentioned in the main text (Section 3.2), we produce sorted tables of varying sizes so that each fits into a level of the internal memory hierarchy as follows. Letting *n* be the number of elements in a table, for the computer architecture we are using, the details of the tables we generate are as follows.

- **Fitting in L1 cache: cache size 64 Kb.** Therefore, we choose $n = 3.7K$. For each dataset, the table corresponding to this type is denoted with the prefix **L1**, e.g., **L1_amzn**, when needed. For each dataset, in order to obtain a *CDF* that resembles one of the original tables, we proceed as follows. Concentrating on **amzn**, since for the other datasets the procedure is analogous, we extract uniformly and at random a sample of the data of the required size. For each sample, we compute its *CDF*. Then, we use the Kolmogorov–Smirnov test [57] in order to assess whether the *CDF* of the sample is different than the **amzn** *CDF*.
  If the test returns that we cannot exclude such a possibility, we compute the PDF of the sample and compute its KL divergence [58] from the PDF of **amzn**. We repeat such a process 100 times and, for our experiments, we use the sample dataset with the smallest KL divergence.
- **Fitting in L2 cache: cache size 256 Kb.** Therefore, we choose $n = 31.5K$. For each dataset, the table corresponding to this type is denoted with the prefix **L2**, when needed. For each dataset, the generation procedure is the same as the one of the **L1** dataset.
- **Fitting in L3 cache: cache size 8 Mb.** Therefore, we choose $n = 750K$. For each dataset, the table corresponding to this type is denoted with the prefix **L3**, when needed. For each dataset, the generation procedure is the same as the one of the **L1** dataset.

- **Fitting in PC main memory: memory size 32 Gb.** Therefore, we choose $n = 200M$, i.e., the entire dataset. For each dataset, the table corresponding to this type is denoted with the prefix **L4**.

For completeness, the results of the Kolmogorov–Smirnov Test as well as KL divergence computation are reported in Table A1. For each memory level (first row) and each original dataset (first column), we report the percentage of times in which the Kolmogorov–Smirnov test failed to report a difference between the *CDF*s of the dataset extracted uniformly and at random from the original one and this latter, over 100 extractions. Moreover, the KL divergence between the PDFs of the chosen generated dataset and the original one is also reported. From these results, it is evident that the PDF of the original datasets is quite close to the one of the extracted datasets.

**Table A1. Results of the Kolmogorov–Smirnov test**. For each memory level and each dataset, the results of the Kolmogorov–Smirnov test (**%succ** columns) and of the KL divergence computation (**KLdiv** columns) are reported.

| | L1 | | L2 | | L3 | |
|---|---|---|---|---|---|---|
| **Datasets** | **%succ** | **KLdiv** | **%succ** | **KLdiv** | **%succ** | **KLdiv** |
| **amzn** | 100 | $9.54 \times 10^{-6} \pm 7.27 \times 10^{-14}$ | 100 | $7.88 \times 10^{-5} \pm 7.97 \times 10^{-13}$ | 100 | $1.88 \times 10^{-3} \pm 1.52 \times 10^{-11}$ |
| **face** | 100 | $1.98 \times 10^{-5} \pm 1.00 \times 10^{-12}$ | 100 | $7.98 \times 10^{-5} \pm 4.43 \times 10^{-13}$ | 100 | $1.88 \times 10^{-3} \pm 1.24 \times 10^{-11}$ |
| **osm** | 100 | $9.38 \times 10^{-6} \pm 4.51 \times 10^{-14}$ | 100 | $7.88 \times 10^{-5} \pm 3.46 \times 10^{-13}$ | 100 | $1.88 \times 10^{-3} \pm 9.55 \times 10^{-12}$ |
| **wiki** | 100 | $9.47 \times 10^{-6} \pm 5.27 \times 10^{-14}$ | 100 | $7.87 \times 10^{-5} \pm 5.64 \times 10^{-13}$ | 100 | $1.88 \times 10^{-3} \pm 1.25 \times 10^{-11}$ |

*Appendix A.3. Training of the Novel Models: Analysis and Insights into Model Training—Additional Results*

Following the same approach as used in Section 4 of the main text, we divide the training time analysis into two groups: constant- and parametric-space models.

- Tables A2–A4 report the experiments concerning the constant-space models for the datasets **L1**, **L2** and **L3**.
- Tables A5–A7 report the experiments concerning the parametric-space models for the datasets **L1**, **L2** and **L3**.

**Table A2. Constant-space model training time for L1 Tables**. The first column indicates the datasets. The remaining columns indicate the models used for the learning phase. Each entry reports the training time in seconds and per element.

| | **KO-US** | **C** |
|---|---|---|
| **amzn** | $5.3 \times 10^{-7}$ | $1.0 \times 10^{-7}$ |
| **face** | $5.5 \times 10^{-7}$ | $8.5 \times 10^{-8}$ |
| **osm** | $4.6 \times 10^{-7}$ | $9.9 \times 10^{-8}$ |
| **wiki** | $9.0 \times 10^{-7}$ | $7.9 \times 10^{-8}$ |

**Table A3. Constant-space model training time for L2 Tables**. The table legend is as in Table A2.

|      | KO-BFS | C |
|------|--------|---|
| **amzn** | $1.8 \times 10^{-7}$ | $3.1 \times 10^{-7}$ |
| **face** | $1.0 \times 10^{-7}$ | $2.8 \times 10^{-7}$ |
| **osm** | $1.2 \times 10^{-7}$ | $2.9 \times 10^{-7}$ |
| **wiki** | $1.0 \times 10^{-7}$ | $2.7 \times 10^{-7}$ |

**Table A4. Constant-space model training time for L3 Tables**. The table legend is as in Table A2.

|      | KO-US | C |
|------|-------|---|
| **amzn** | $6.3 \times 10^{-8}$ | $1.9 \times 10^{-8}$ |
| **face** | $3.9 \times 10^{-8}$ | $1.9 \times 10^{-8}$ |
| **osm** | $4.4 \times 10^{-8}$ | $2.0 \times 10^{-8}$ |
| **wiki** | $4.1 \times 10^{-8}$ | $1.9 \times 10^{-8}$ |

**Table A5. Parametric-space model training time for L1 Tables**. The first column indicates the datasets. The remaining columns indicate the models used for the learning phase. In particular, each entry reports the time used by the **CDFShop** and **SOSD** libraries to train the entire batch of parametric models in seconds and per element.

|      | CDFShop SY-RMI 2% | CDFShop RMI | SOSD RS | SOSD PGM |
|------|-------------------|-------------|---------|----------|
| **amzn** | $5.2 \times 10^{-6}$ | $5.6 \times 10^{-6}$ | $3.5 \times 10^{-6}$ | $5.0 \times 10^{-7}$ |
| **face** | $4.1 \times 10^{-6}$ | $4.6 \times 10^{-6}$ | $1.1 \times 10^{-6}$ | $3.9 \times 10^{-7}$ |
| **osm** | $2.8 \times 10^{-4}$ | $2.9 \times 10^{-4}$ | $6.9 \times 10^{-6}$ | $4.0 \times 10^{-7}$ |
| **wiki** | $7.8 \times 10^{-6}$ | $9.3 \times 10^{-6}$ | $1.0 \times 10^{-5}$ | $3.7 \times 10^{-7}$ |

**Table A6. Parametric-space model training time for L2 Tables**. The table legend is as in Table A5.

|      | CDFShop SY-RMI 2% | CDFShop RMI | SOSD RS | SOSD PGM |
|------|-------------------|-------------|---------|----------|
| **amzn** | $5.2 \times 10^{-6}$ | $5.6 \times 10^{-7}$ | $3.5 \times 10^{-7}$ | $5.0 \times 10^{-8}$ |
| **face** | $4.1 \times 10^{-6}$ | $4.6 \times 10^{-7}$ | $1.1 \times 10^{-7}$ | $3.9 \times 10^{-8}$ |
| **osm** | $2.8 \times 10^{-4}$ | $2.9 \times 10^{-5}$ | $6.9 \times 10^{-7}$ | $4.0 \times 10^{-8}$ |
| **wiki** | $7.8 \times 10^{-6}$ | $9.3 \times 10^{-7}$ | $1.0 \times 10^{-6}$ | $3.7 \times 10^{-8}$ |

**Table A7. Parametric-space model training time for L3 Tables**. The table legend is as in Table A5.

|      | CDFShop SY-RMI 2% | CDFShop RMI | SOSD RS | SOSD PGM |
|------|-------------------|-------------|---------|----------|
| **amzn** | $1.5 \times 10^{-6}$ | $1.3 \times 10^{-7}$ | $2.4 \times 10^{-8}$ | $3.4 \times 10^{-8}$ |
| **face** | $1.5 \times 10^{-5}$ | $1.6 \times 10^{-6}$ | $1.4 \times 10^{-8}$ | $2.4 \times 10^{-8}$ |
| **osm** | $1.2 \times 10^{-5}$ | $1.3 \times 10^{-6}$ | $3.5 \times 10^{-8}$ | $3.8 \times 10^{-8}$ |
| **wiki** | $2.3 \times 10^{-6}$ | $2.2 \times 10^{-7}$ | $5.1 \times 10^{-8}$ | $3.7 \times 10^{-8}$ |

*Appendix A.4. Accuracy of Query Time Evaluation*

With reference to Section 3.2 of the main text, we provide a highlight that the processing time for a batch of queries, over independent executions, is stable. In particular, we concentrate on the **amzn** datasets and on the Learned *k*-are Search, the Synoptic **RMI**, the **RMI**, the **PGM** and the **RS** models. A query batch of 2 million elements, obtained as specified in Section 3.2 of the main text, is processed 10 times. In Table A8, we report the average batch-query time processing (with the standard deviation), which is indeed low.

**Table A8. Batch-query time processing over independent executions.** The first row indicates the model, while the memory level is indicated by the first column. Each entry in the table indicates the average time (in seconds) to execute 10 times the same batch of queries together with the corresponding standard deviation.

| | RMI | PGM | RS | SY-RMI 0.05 | KO-US |
|---|---|---|---|---|---|
| **L1** | $2.35 \times 10^{-2} \pm$ $5.56 \times 10^{-4}$ | $3.27 \times 10^{-2} \pm$ $1.49 \times 10^{-3}$ | $2.61 \times 10^{-2} \pm$ $3.69 \times 10^{-4}$ | $2.34 \times 10^{-2} \pm$ $3.69 \times 10^{-4}$ | $2.90 \times 10^{-1} \pm$ $1.1 \times 10^{-2}$ |
| **L2** | $3.02 \times 10^{-2} \pm$ $4.20 \times 10^{-4}$ | $3.88 \times 10^{-2} \pm$ $9.93 \times 10^{-4}$ | $3.44 \times 10^{-2} \pm$ $2.89 \times 10^{-4}$ | $3.02 \times 10^{-2} \pm$ $4.20 \times 10^{-4}$ | $2.99 \times 10^{-1} \pm$ $8.04 \times 10^{-3}$ |
| **L3** | $6.78 \times 10^{-2} \pm$ $1.02 \times 10^{-3}$ | $7.16 \times 10^{-2} \pm$ $1.31 \times 10^{-3}$ | $8.33 \times 10^{-2} \pm$ $1.40 \times 10^{-3}$ | $6.78 \times 10^{-2} \pm$ $1.02 \times 10^{-3}$ | $3.81 \times 10^{-1} \pm$ $1.09 \times 10^{-2}$ |
| **L4** | $1.66 \times 10^{-1} \pm$ $3.93 \times 10^{-3}$ | $1.67 \times 10^{-1} \pm$ $1.90 \times 10^{-3}$ | $1.62 \times 10^{-1} \pm$ $1.53 \times 10^{-3}$ | $1.66 \times 10^{-1} \pm$ $3.94 \times 10^{-3}$ | $7.37 \times 10^{-1} \pm$ $5.40 \times 10^{-3}$ |

*Appendix A.5. Query Experiments—Additional Results*

In this section, we report the experiments described and discussed in Section 5 of the main text for the **face** and **wiki** datasets.

- Figures A2 and A3 report the experiments concerning the constant-space models as in Section 5.1.
- Figures A4 and A5 report the experiments concerning the parametric-space models as in Section 5.2.
- Tables A9 and A10 report a synopsis of three parameters, i.e., the query time, space used in addition by the model and reduction factor as described in Section 5.2.

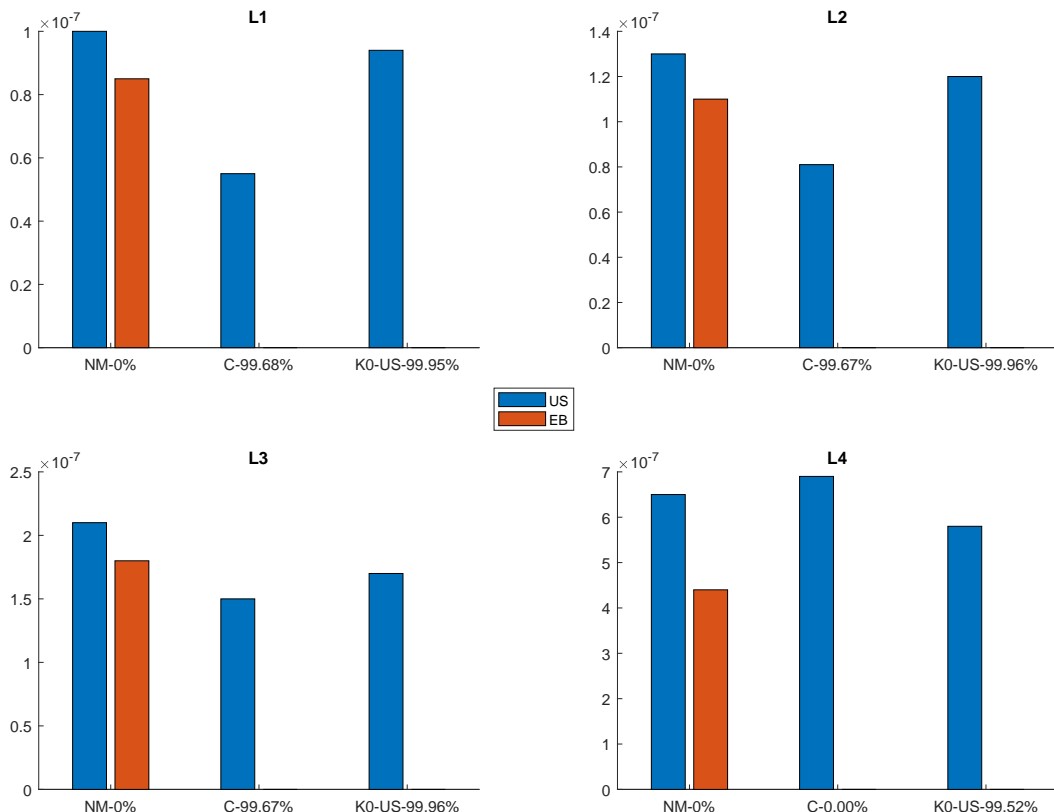

**Figure A2. Constant-space-model query times for the face dataset**. For each memory level, the blue bar reports the average query time in seconds of Uniform Binary Search using, from left to right, no model, the Cubic model and **KO-US**. In addition, we also report the average query time for the Eytzinger Binary Search in the orange bar.

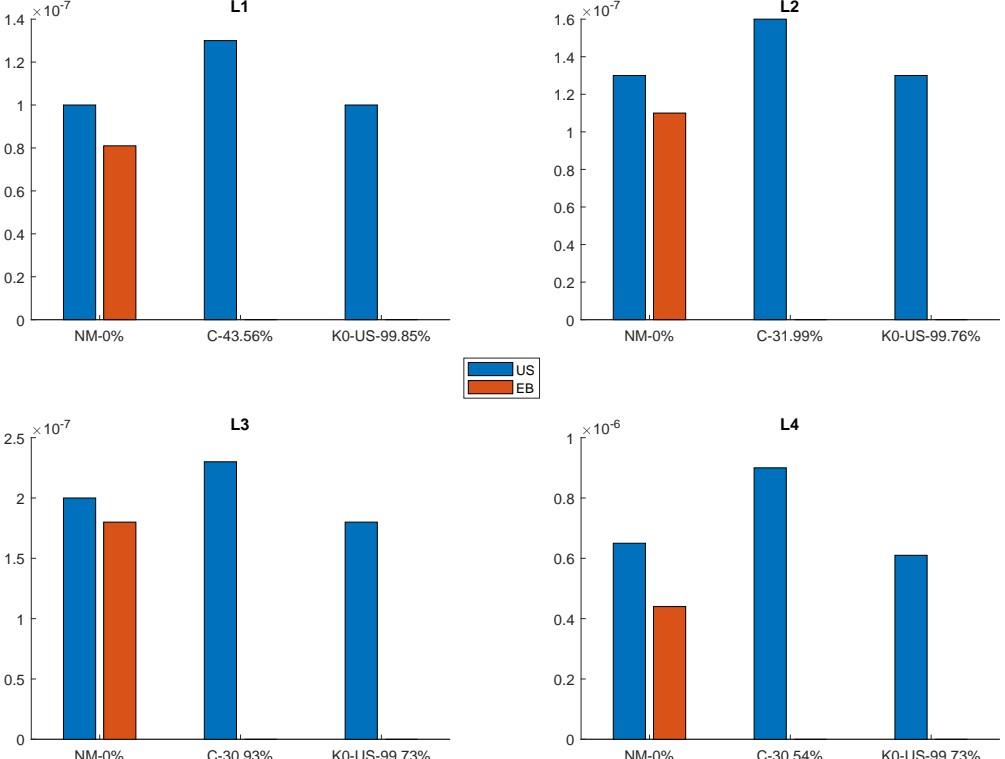

**Figure A3. Constant-space-model query times for the wiki dataset**. The legend is as in Figure A2.

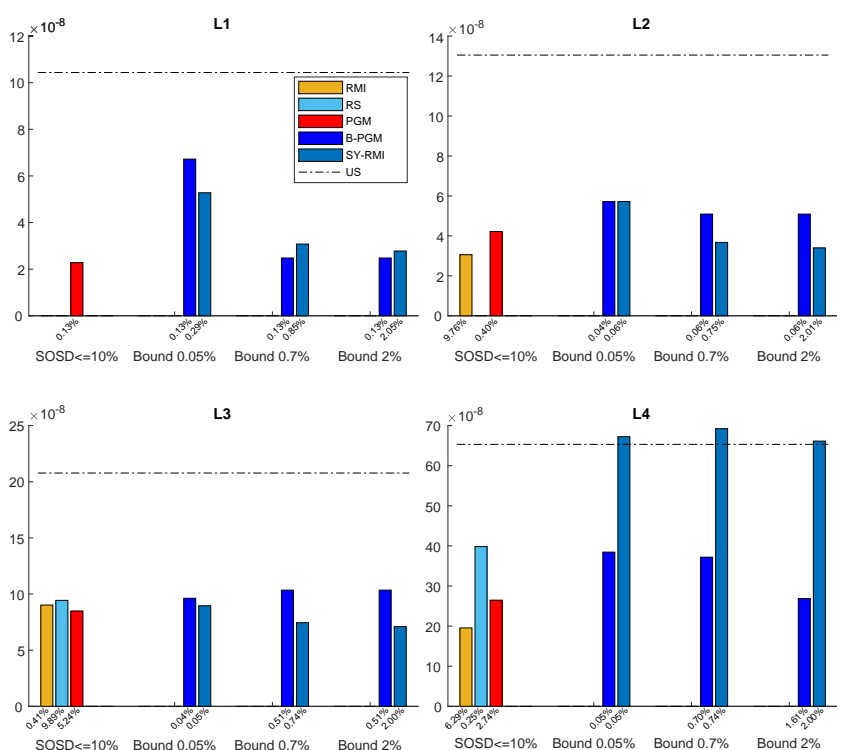

**Figure A4. Query times for the face dataset on Learned Indexes in small space.** The methods are the ones in the legend (middle of the four panels, the notation is as in the main text and each method has a distinct colour). For each memory level, the abscissa reports methods grouped by space occupancy, as specified in the main text. When no model in a class output by **SOSD** takes at most 10% of additional space, that class is absent. The ordinate reports the average query time in seconds, with Uniform Binary Search executed in **SOSD** as baseline (horizontal lines).

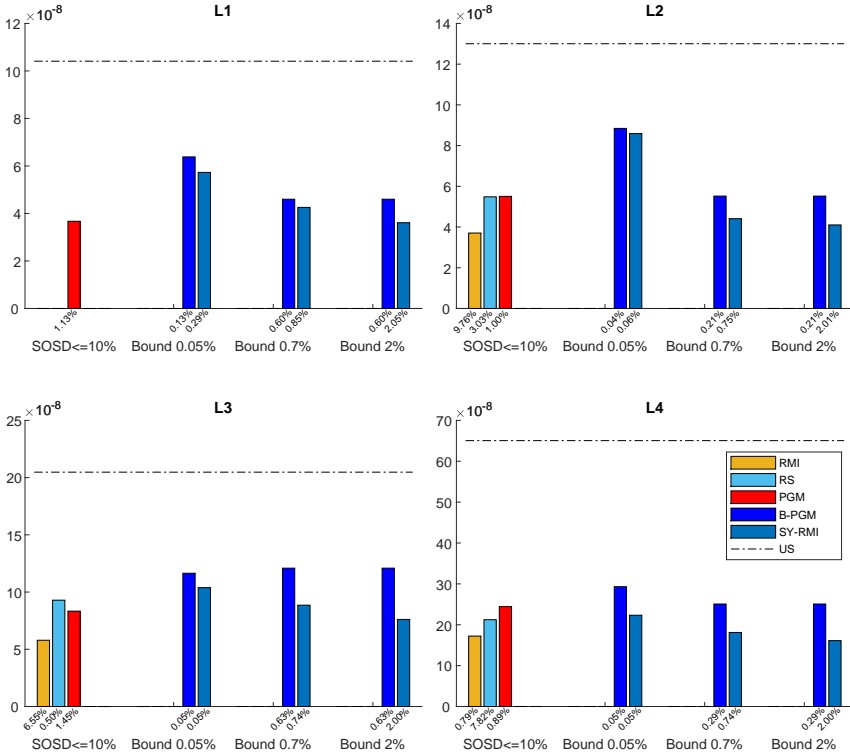

**Figure A5. Query times for the wiki dataset on Learned Indexes in small space.** The figure legend is as in Figure A4.

**Table A9. A synoptic table of space, time and accuracy of the models on the face dataset**. For each memory level, we report, in the first row, the best performing method for that memory level. The columns named time, space and reduction factor indicate, for this best model, the average query time in seconds, the average additional space used and the average of the empirical reduction factor. From the second row, we report the versions of the **RMI**, **RS**, **PGM** and Synoptic **RMI** models that use the least space. In particular, the numbers next to the models represent the percentage of the used space with respect to the input dataset. The columns indicate the ratio of the model/best model of the time, space and reduction factor.

| | | L1 | |
| --- | --- | --- | --- |
| | **Time** | **Space** | **Reduction Factor** |
| Best **PGM** | $2.26 \times 10^{-8}$ | $4.00 \times 10^{-2}$ | 99.52 |
| Best **PGM** | 1.00 | 1.00 | 1.00 |
| **SY-RMI** 0.05 | 2.33 | 2.20 | $6.30 \times 10^{-1}$ |
| Best **RMI** | 1.16 | $7.72 \times 10^{1}$ | 1.00 |
| Best **RS** | 1.17 | $5.90 \times 10^{4}$ | 1.00 |
| | | L2 | |
| | **Time** | **Space** | **Reduction Factor** |
| Best **RMI** | $3.02 \times 10^{-8}$ | $1.23 \times 10$ | 99.98 |
| **B-PGM** 0.05 | 1.89 | $8.13 \times 10^{-3}$ | $9.98 \times 10^{-1}$ |
| **SY-RMI** 0.05 | 1.89 | $1.30 \times 10^{-2}$ | $9.40 \times 10^{-1}$ |
| Best **RMI** | 1.00 | 1.00 | 1.00 |
| Best **RS** | 1.10 | $1.92 \times 10^{2}$ | 1.00 |
| | | L3 | |
| | **Time** | **Space** | **Reduction Factor** |
| Best **RMI** | $6.11 \times 10^{-8}$ | $7.86 \times 10^{2}$ | 100.00 |
| **B-PGM** 0.05 | 1.57 | $3.33 \times 10^{-3}$ | 1.00 |
| **RMI** $< 10$ | 1.19 | $3.13 \times 10^{-2}$ | 1.00 |
| **SY-RMI** 0.7 | 1.22 | $5.62 \times \times 10^{-2}$ | 1.00 |
| **RS** $< 10$ | 1.53 | $7.54 \times 10^{-1}$ | 1.00 |
| | | L4 | |
| | Time | Space | Reduction Factor |
| Best **RMI** | $1.80 \times 10^{-7}$ | $2.01 \times 10^{5}$ | 100.00 |
| **SY-RMI** 0.05 | 3.74 | $3.97 \times 10^{-3}$ | $1.32 \times 10^{-2}$ |
| **B-PGM** 0.05 | 2.14 | $3.98 \times 10^{-3}$ | 1.00 |
| Best **RS** | 2.21 | $3.96 \times 10^{-2}$ | 1.00 |
| **RMI** $< 10$ | 1.06 | $5.00 \times 10^{-1}$ | 1.00 |

**Table A10. A synoptic table of space, time and accuracy of the models on the wiki dataset**. The legend is as in Table A9.

| | Time | Space | Reduction Factor |
|---|---|---|---|
| **L1** | | | |
| | **Time** | **Space** | **Reduction Factor** |
| Best **RMI** | $2.55 \times 10^{-8}$ | 3.09 | 99.84 |
| **B-PGM** 0.05 | 2.50 | $1.30 \times 10^{-2}$ | $2.06 \times 10^{-1}$ |
| **SY-RMI** 0.05 | 2.24 | $2.85 \times 10^{-2}$ | $8.52 \times 10^{-1}$ |
| Best **RMI** | 1.00 | 1.00 | 1.00 |
| Best **RS** | 1.70 | 2.40 | $9.77 \times 10^{-1}$ |
| **L2** | | | |
| | **Time** | **Space** | **Reduction Factor** |
| Best **RMI** | $3.32 \times 10^{-8}$ | $9.83 \times 10^{1}$ | 99.98 |
| **B-PGM** 0.05 | 2.66 | $1.02 \times 10^{-3}$ | $9.26 \times 10^{-1}$ |
| **SY-RMI** 0.05 | 2.59 | $1.63 \times 10^{-3}$ | $9.57 \times 10^{-1}$ |
| Best **RS** | 1.60 | $7.77 \times 10^{-2}$ | $9.97 \times 10^{-1}$ |
| **RMI** $< 10$ | 1.05 | $2.50 \times 10^{-1}$ | 1.00 |
| **L3** | | | |
| | **Time** | **Space** | **Reduction Factor** |
| Best **RMI** | $5.16 \times 10^{-8}$ | $7.86 \times 10^{2}$ | 100.00 |
| **B-PGM** 0.05 | 2.26 | $3.76 \times 10^{-3}$ | 1.00 |
| **SY-RMI** 0.05 | 2.01 | $3.83 \times 10^{-3}$ | 1.00 |
| Best **RS** | 1.74 | $3.82 \times 10^{-2}$ | 1.00 |
| **RMI** $< 10$ | 1.14 | $5.00 \times 10^{-1}$ | 1.00 |
| **L4** | | | |
| | **Time** | **Space** | **Reduction Factor** |
| **SY-RMI** 2 | $1.61 \times 10^{-7}$ | $3.20 \times 10^{4}$ | 100.00 |
| **SY-RMI** 0.05 | 1.39 | $2.50 \times 10^{-2}$ | 1.00 |
| **B-PGM** 0.05 | 1.82 | $2.53 \times 10^{-2}$ | 1.00 |
| Best **RS** | 1.30 | $4.97 \times 10^{-1}$ | 1.00 |
| **RMI** $< 10$ | 1.02 | $7.86 \times 10^{-1}$ | 1.00 |

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
