# Peer review of "Learned Sorted Table Search and Static Indexes in Small-Space Data Models†"

_data, 2023_

Round 1

Reviewer 1 Report

This is a well-written paper with some useful findings. However, the results need to be put in more context.

Generic remarks:

- The introduction starts with terminology such as 'Data Structures'. These should be explained (briefly).

- It would be good to know some real-life examples in which the reported findings would be useful. The aim of the "Data" journal is to "enhance data transparency and reusability". Explain how this paper contributes to that aim. Since there is a time-space trade/off, for which examples would "time" be more important and for which "space"?

- I don't think the Appendix Tables/Figures should be mixed with the references?

Typos:

- line 53: "However, The"-> "However, the"

- line 142: "previuos" -> "previous", "fuil"-> "full"

- table 2: "Paramentric Models" -> "Parametric Models"

- line 641: "version" -> "versions"

Reviewer 2 Report

This paper introduces 2 models (the Learned k-ary Search Model and the Synoptic Recursive Model Index) to assess the Binary Search performance achieved by Learned Indexes in constant or nearly constant space models. Based on the experimental results, the Learned k-ary Search Model can increase the speed of Binary Search in constant additional space and the second model can increase Binary Search by 0.05% compared to taken by the table. Peper is well-presented and easy to read. However, it needs confirmation and a comment:

1. What is the author's main problem, why is it important to assess to what extent one can enjoy the Binary Search acceleration achieved by Learned Index? The author needs to disclose data analysis on the main problem.

2. The author needs to explain the research roadmap, including previous research that has been done, such as the position of the following article: https://link.springer.com/chapter/10.1007/978-3-031-08421-8_32

3. Authors need to cite previous articles that propose KO-BFS and SY-RMI.

4. The conclusions are slightly inconsistent with the evidence and arguments presented. Kindly check if the main questions asked are fully answered.

The conclusions are slightly inconsistent with the evidence and arguments presented. Kindly check if the main questions asked are fully answered.

5. Some of the text in tables and figures is too small, and hard to read.

6. The author needs to discuss a little about how the sampling and testing scenarios are carried out.

7. In addition, the accuracy of data processing is important to be explored.

Round 2

Reviewer 2 Report

This article has been properly revised, there may only be a few grammatical errors, and it is better if the appendix is placed after the reference.

Author Response

Thank you for the appreciation of the work we have done in revising the manuscript. In this new version, we have moved the appendix as requested